



# Evaluation of correlated Pandora column observations and *in situ* surface air quality measurements during GMAP campaign

Lim-Seok Chang[1], Donghee Kim[1], Hyunkee Hong[1], Deok-Rae Kim[1], Jeonga Yu[1],

Kwangyul Lee[1], Hanlim Lee[2], Daewon Kim[2], Jinkyu Hong[3], Hyun-Young Jo[4],

and Cheol-Hee Kim[4, 5]

[1] *Environmental Satellite Center, National Institute of Environmental Research, Incheon, 22689, Republic of Korea*

[2] *Department of Spatial Information Engineering, Pukyong National University, Busan, 48547, Republic of Korea*

[3] *Department of Atmospheric Sciences, Yonsei University, Seoul, 03722, Republic of Korea*

[4] *Institute of Environmental Studies, Pusan National University, Busan, 46241, Republic of Korea*

[5] *Department of Atmospheric Sciences, Pusan National University, Busan, 46241, Republic of Korea*

-------------------------------------------------------------------------------------------------------------

**Correspondence**:
Cheol-Hee Kim (chkim2@pusan.ac.kr), and Lim-Seok Chang (lschang@korea.kr)



**Abstract.**
To validate the Geostationary Environment Monitoring Spectrometer (GEMS), the GEMS
Map of Air Pollution (GMAP) campaign was conducted during 2020–2021 by integrating
Pandora Asia Network, aircraft, and *in situ* measurements. In the present study, GMAP-2020
measurements were applied to evaluate urban air quality and explore the synergy of Pandora
column (PC) $NO_2$ measurements and surface *in situ* (SI) $NO_2$ measurements for Seosan, South
Korea, where large point source (LPS) emissions are densely clustered. Due to the difficulty
of interpreting the effects of LPS emissions on air quality downwind of Seosan using SI
monitoring networks alone, we used a combination of PC and SI measurements, and explored
the synergy of this approach through correlation analysis of PC-$NO_2$ and SI-$NO_2$.
Agglomerative hierarchical clustering using vertical meteorological variables combined with
PC-$NO_2$ and SI-$NO_2$ yielded three distinct conditions: synoptic wind-dominant (SD), mixed
(MD), and local wind-dominant (LD). These results suggested meteorology-dependent
correlations between PC-$NO_2$ and SI-$NO_2$. Overall, yearly daytime mean (11:00–17:00 KST)
PC-$NO_2$ and SI-$NO_2$ statistical data showed good linear correlations ($R$ = ~0.73); however,
these correlations were dependent on meteorological conditions. SD conditions characterized
by higher wind speeds and planetary boundary layer heights suppressed fluctuations in both
PC-$NO_2$ and SI-$NO_2$, driving a uniform vertical $NO_2$ structure with higher correlations,
whereas under LD conditions, stack plumes decoupled from LPS or were transported from
nearby cities, weakening correlations through anomalous vertical $NO_2$ gradients. However,
under MD conditions, both pollution ventilation due to high surface wind speeds and daytime
photochemical $NO_2$ loss contributed to stronger correlations through a decline in both PC-$NO_2$
and SI-$NO_2$ toward noon. Thus, Pandora Asia Network observations collected over 13 Asian
countries since 2021 can be utilized for investigation of the vertical complexity of air quality
in combination with SI measurements. The results of this study also indicate that caution is
required when performing GEMS validation using either PC or SI observations alone,
particularly under prevailing local wind meteorological conditions or transport processes.
**1. Introduction**
Rapid developments in environmental remote sensing have led to a new era of air quality
observations, and recent hyperspectral data retrieval technologies have allowed for routine and




accurate monitoring of air pollutants at high spatial and temporal resolution. In particular, the
Geostationary Environment Monitoring Spectrometer (GEMS), which was launched on
February 18, 2020, measures the total and tropospheric air pollutant columns hourly at spatial
resolutions of 7 km × 8 km for gas and 3.5 km × 8 km for aerosols (Kim and Kim, 2020),
facilitating the tracking of pollution transport from local to synoptic scales.

Recent studies have revealed the potential of satellite observations to evaluate surface air

quality, particularly in regions with sparse air quality monitoring networks. The main approach
is to convert column amounts to surface concentrations using a shape factor of the ratio of the
partial column ($\Omega_{z_0}$) within the lowest layer ($z_0$) to the total column ($\Omega_{total}$), as follows:

$$S = \frac{\Omega_{z_0}}{\Omega_{total}} \times \frac{C}{\Delta z}$$

where S, C and $\Delta z$ are the surface concentration, column amount, and thickness of the lowest
layer, respectively. Because the shape factor is spatially and temporally variable, it is obtained
through the simulation of chemical transport model or aircraft *in situ* measurements. Acquiring
accurate profile shape information is critical for determining the close relationship between the
column amount and surface concentration. Lamsal et al. (2010) obtained a good correlation
between *in situ* surface $NO_2$ and Ozone Monitoring Instrument (OMI)-derived surface $NO_2$ by
applying local shape factors from the GEOS-Chem model, because the vertical $NO_2$ profile
calculated by GEOS-Chem is consistent with *in situ* aircraft measurements. Other studies have
instead assumed a uniform vertical profile to convert column amounts to surface
concentrations. Wang and Christopher (2003) found a linear relationship between Moderate
Resolution Imaging Spectroradiometer (MODIS) aerosol optical depth (AOD) and surface fine
particulate matter ($PM_{2.5}$) in Alabama, USA, with a high  correlation coefficient (*R*) of 0.7.
This strong correlation can be explained by a generally uniform planetary boundary layer
height (PBLH), as well as by aerosol type and abundance, which is also the case for trace gases.



A uniform vertical PBL profile has also been used to successfully scale up surface $NO_2$ to
column $NO_2$ in Israeli cities, producing results consistent with the OMI column $NO_2$ (Boersma
et al., 2009).

By contrast, the implications of weak vertical profile correlations remain unclear. Engel-

Cox et al. (2004) found a negative correlation of AOD and surface $PM_{2.5}$ in northwestern USA,
and explained it based on elevated haze decoupled from the surface. Thompson et al. (2019)
examined weak correlations between Pandora column (PC) measurements and surface *in situ*
(SI) observations of $NO_2$ over the Yellow Sea during the Korea–US air quality (KORUS-AQ)
field study, and found that they originated from plumes in China and Seoul hundreds of meters
above the ground (detached from the surface layer). The estimated surface $PM_{2.5}$ concentration
was weakly correlated ($R = 0.4$–$0.49$) with observed $PM_{2.5}$ concentrations in Seoul, because
only PBLH was added to the multi-linear regression model to correlate AOD to surface $PM_{2.5}$
(Kim et al., 2021). This effect may be related to the significant impact of long-range transport
on $PM_{2.5}$, with a contribution of up to 39% in Seoul (Lee et al, 2021). Thus, the wide variability
in the degree of correlation between PC-PM and SI-PM is closely related to vertical profile
variability (Flynn et al., 2016).

It appears highly probable that several factors are responsible for the correlations between

PC-$NO_2$ and SI-$NO_2$; therefore, it is necessary to improve our understanding of the degree of
correlation through detailed measurements, including column concentration. In this study, we
focused on the impact of meteorology and chemistry on correlation variability using PC, SI
and aircraft measurements, as well as meteorological observations. Understanding vertical
profile variability is also useful for evaluating the effects of various emissions on urban air
quality, particularly in areas neighboring active large point source (LPS) emission sites.
Quantifying the impact of LPS emissions on downwind cities remains challenging due to the
lack of three-dimensional (3D) measurements. Accurate vertical profile data are also useful for



improving remote sensing retrieval algorithms, because the profile shape contributes to the
conversion of slant column density into vertical column density as part of the air mass factor.
In mid-2019, the Pandonia Global Network (PGN; https://pandonia-global-network.org)
was launched, with support from the National Aeronautics and Space Administration (NASA)
and European Space Agency (ESA), to facilitate the validation and verification of low-orbit
geostationary environmental satellites. This network is attempting to expand air quality
monitoring through integration with existing long-term air quality monitoring stations. Since
2020, the National Institute of Environmental Research, Economic and Social Commission for
Asia and the Pacific, and Korea Environment Corporation have been extending the Pandora
Asia Network to include 13 Asian countries, with support from the Korea International
Cooperation Agency. The Pandora Asia Network is expected to be widely used to study urban
air quality in Asia, which is increasingly deteriorating due to rapid economic growth.
As part of the GEMS Map of Air Pollution (GMAP) campaign, a suite of Pandora
instruments was deployed in Seosan, a coastal city of South Korea, from November 2020 to
January 2021 (GMAP-2020), and in the Seoul metropolitan area from October 2021 to
November 2021 (GMAP-2021). In this study, we applied GMAP-2020 measurements to
explore the synergy of PC observations when evaluating air quality over Seosan. Further results
from this research project are also reported in this special issue, including GEMS validation
and urban air quality evaluations based on Pandora, aircraft, surface flux, and *in situ* surface
chemical measurements conducted during GMAP-2020 and GMAP-2021.

**2. GMAP campaign**
GEMS was launched on February 19, 2020; it is the first instrument to observe air quality from
a geostationary Earth orbit. GEMS provides hourly air quality data on aerosols and gases at a
spatial resolution of 7 km × 8 km. GEMS is a scanning ultraviolet (UV)–visible spectrometer


that observes key atmospheric constituents including $O_3$, $NO_2$, CO, $SO_2$, $CH_2O$, CHOCHO,
aerosols, clouds, and UV indices. This mission heralded a new era of satellite air quality
monitoring and will be joined by NASA's Tropospheric Emissions: Monitoring of Pollution
(TEMPO) and ESA's Sentinel-4 to form the GEO Air Quality Constellation in ~3 years, to
cover the most polluted region in the Northern Hemisphere.
During GMAP-2020, Pandora instruments were deployed near LPSs in Seosan. Aircraft
measurements and *in situ* surface air quality monitoring systems were used to validate GEMS
and diagnose LPSs located in industrial areas surrounding Seosan. During GMAP-2021,
differential optical absorption spectroscopy (DOAS), car DOAS (Car-DOAS), aircraft, and
Geo-CAPE airborne simulator (GCAS) measurements were also used to validate and evaluate
air quality over the Seoul metropolitan area. In this study, we explored the synergy of Pandora
observations and *in situ* surface measurements, based on measurements collected during
GMAP-2020, by evaluating air quality in industrial Seosan (where LPSs are densely clustered).
During GMAP-2021, multi-perspective observations were obtained from the ground, air, and
space; participating remote sensing instruments were expanded to include multi-axis (Max)
DOAS, Car-DOAS, GCAS, and Pandora data. The target area was the Seoul metropolitan area
and target pollutants included $O_3$, HCHO, $SO_2$, aerosols, and $NO_2$. We investigated the impacts
of vertical profile and sub-pixel variability for trace gases and aerosols, for further GEMS
validation. All measurement sites for both GMAP campaigns are indicated in Figure 1.

**3. Methods**
**3.1 Study area**
Seosan, the target area of the GMAP-2020 campaign, is a small city with a population of
174,780 in 2017; it is accessed via three expressways to the east and four national highways
cross the city. Seosan is located in midwestern South Korea, and is affected by > 300 emission





point sources including LPSs. Coal-fired power plants including Taean, Dangjin, the Hyundai
Dangjin steelworks, and the Daesan petrochemistry industrial complex ($LPS_1$–$LPS_4$,
respectively, in Fig. 1) have the highest emission rates in South Korea. The Hyundai Dangjin
steelworks ($LPS_3$) and Taean and Dangjin power plants ($LPS_1$ and $LPS_2$) emit 10.5, 11, and
8.8 Gg of NOx per year, respectively. Although Seosan accounts for only 1.8% of the
population of Seoul, its NOx emissions (10.2 Gg $year^{-1}$) account for 13.2% of its total NOx
emissions. The transportation sector of Seosan is a far greater NOx source than the industrial
sector of Seoul (ratio of 99:1); however, within Seosan, the industrial sector is on par with the
transport sector (52:48; http://airemiss.nier.go.kr).
During the past decade, the annual mean $NO_2$ level in Seosan has been 17 ppb, which is
approximately half of that in Seoul (31.2 ppb). $NO_2$ exhibits strong seasonal variation, reaching
a minimum in summer and maximum in winter, due to meteorological factors and greater
energy use during winter (Kim and Kim, 2020). Therefore, the timing of the GMAP-2020
campaign was well suited to tracking pollution.

**3.2 Pandora measurements**
Pandora measures the UV and visible wavelengths (280–525 nm) of direct sunlight with a
spectral resolution of 0.6 nm, to determine the vertical column density of $NO_2$, $O_3$, and HCHO
(Herman et al., 2009). For measurements in Dobson units (DU; 1 DU = 26.9 Pmol $cm^{-2}$),
column $NO_2$ has a very high signal-to-noise ratio (700:1) and very high precision (0.01 DU)
for clear skies (Herman et al., 2009). The vertical column density of $NO_2$ can be determined
using DOAS software (Van Roozendael and Fayt, 2001). Pandora direct-sun measurements are
advantageous in that the air mass factor is simplified, and is therefore dependent only on the
geography for a known solar zenith angle.
Four Pandora instruments were installed at sites to the south of LPSs (Fig. 1) during the



GMAP-2020 campaign, i.e., at Seosan Daehoji, Seosan Dongmun, Seosan City Council, and
Seosan Super Site ($PA_1$–$PA_4$ in Fig. 1). The presence of clouds reduces vertical column density
precision by decreasing the number of photons arriving at Pandora instruments within a fixed
integration time. Therefore, the retrieved Pandora measurements were cloud-screened using an
observed cloud cover of 0.6. Cloud cover was provided by the Korea Meteorological
Administration (KMA), and the precision improvement afforded by cloud screening was
verified by comparing each Pandora-derived vertical column density with the median vertical
column density, with and without cloud screening within the inter-comparison period.

At $PA_4$, the operating period was extended to cover almost the entire year (November 12,

2020–October 30, 2021) including the GMAP-2020 campaign period, and the Pandora spectra
were processed into vertical column density data for trace gases using the standard $NO_2$
algorithm in BlickP software provided by PGN (Cede, 2019). The resultant PC-$NO_2$ data were
obtained from the PGN website (https://pandonia-global-network.org) for the 1-year period
from Nov. 12, 2020 to Oct. 30, 2021, and used as PC-$NO_2$ statistics.

**3.3 Surface and airborne chemical measurements**
Hourly average data for SI-$NO_2$ over a period of 1 year were obtained from Ministry of
Environment AQM network stations in Seosan: Pandori, Leewon, Taean, Dongmoon,
Seongyeon, and Daesan ($AQM_1$–$AQM_6$, respectively, in Fig. 1). The Seosan Super Site
($PA_4/AQM_1$) provided hourly data for NO and $NO_y$ via an NO-DIF-$NO_y$ analyzer (42i-Y;
Thermo Scientific, Waltham, MA, USA), and for $PM_{2.5}$ chemical species using an ambient ion
monitor (AIM; URG 9000D, URG Corp., Chapel Hill, NC, USA). Weekly zero and span
checks were conducted for $NO_y$ calibration, to ensure that differences between checks
remained < 3%. Water-soluble ions in aerosol and gaseous species were measured hourly using
an AIM, and ion mass balance was used to ensure data quality under the quality control



procedures of the AQM network installation and operation guidelines (NIER-GP2021-002).

Aircraft measurements were conducted during the GMAP-2020 and GMAP-2021

campaign periods. During GMAP-2020, nine flights were conducted on 8 days (Nov. 26, 27,
and 28 and Dec. 1, 6, 8, 9, and 12, 2020). The horizontal and vertical distributions of $NO_2$ and
$O_3$ over Seosan were measured during GMAP-2020 using an $NO_2$ monitor (T500U; Teledyne,
Thousand Oaks, CA, USA) and an $O_3$ analyzer (TEI49C; Thermo Scientific) onboard the
Cessna Grand Caravan 208 B. These instruments had response times of $< 40$ and $< 20$ s, and
detection limits of 40 ppt and 1 ppb, respectively. The flight paths included a raster mode over
all of Seosan at a height of 500–700 m and a profiling mode from 500 m to 1.5 km over $PA_1$
and $PA_4$ (Fig. 2).

**3.4 Meteorological measurements**
Ground-based hourly observation data for meteorological variables were obtained from Seosan
Automated Synoptic Observing System (ASOS) stations maintained by the KMA, and wind
and temperature profile data were obtained twice daily (0000 and 1200 UTC) via a rawinsonde
instrument at the Osan World Meteorological Organization upper air measurement station
(47122) near Seosan. Due to time constraints of the sonde measurements, information on PBLH
variation was obtained from Unified Model (UM) simulation results provided on the KMA
website (https://afso.kma.go.kr).

During the GMAP-2020 campaign, a 3D sonic anemometer (CPEC200; Campbell

Scientific Inc., Logan, UT, USA) was also installed on the rooftop at $PA_4$ for turbulent flux
measurements at the city–atmosphere interface (Hong et al., 2019). All wind components and
sonic temperatures were measured at a 10-Hz sampling rate, and ground-level sensitive heat
flux was measured directly using a 30-min averaging period. Quality controls such as double
rotation, spike removal, and outlier filtering were also applied.






### 3.5 Correlation analyses


The purpose of this study was to examine the synergy of PC and SI data obtained during the
GMAP-2020 campaign, and to combine these measurement data to evaluate air quality in
Seosan, South Korea. We attempted to interpret the meteorological and photochemistry data
measured during GMAP-2020, and to demonstrate that caution is required when attempting to
validate GEMS satellite data through comparison with surface observations only, especially in
industrial areas.
First, we examined the combined use of year-long PC-$NO_2$ and SI-$NO_2$ measurements,
and investigated the factors modulating their correlation. We hypothesized that their
differences were due to meteorological conditions, and performed k-means and agglomerative
hierarchical cluster analyses of meteorological variables using XLSTAT software (Addinsoft
Co., Paris, France). We included eight meteorological variables representing local and synoptic
circulations in the cluster analysis: surface wind speed (Wsfc), 925-hPa temperature (T925),
sea level pressure (Psfc), pressure tendency (dPsfc/dt), 850-hPa wind speed (W850) and its
north–south and east–west components (NS850 and EW850), and 500-hPa geopotential height
(GPH500). We subtracted 30-day moving averages from all data to account for typical seasonal
variation. Monthly averages were used for PC-$NO_2$ analysis due to the limited availability of
hourly data.
Correlations between PC-$NO_2$ and SI-$NO_2$ were analyzed in each meteorological group
and the impact of photochemistry was interpreted based on case-specific features. We also
investigated correlations in association with near-surface micrometeorological variables such
as PBLH in each meteorological group.

### 4. Results and Discussion




## 4.1 Correlation analysis results for PC-NO$_2$ and SI-NO$_2$

The yearly PC-NO$_2$ statistics at four Pandora sites (PA$_1$–PA$_4$) are summarized in Table 1. The
total averaged PC–NO$_2$ over all sites was 0.45 DU during GMAP-2020, which is well above
the typical values (0.1–0.2 DU) for Anmyeondo (location is denoted in Figure 1), a
representative background site (Herman et al., 2018). Although site PA$_3$ is located in a rural
area, it nevertheless exhibited the highest PC-NO$_2$ amounts, suggesting that plumes were
frequently transported from nearby point sources and/or urban areas.
Scatter diagrams of hourly PC-NO$_2$ and SI-NO$_2$ measurements from Pandora sites PA$_1$–
PA$_3$ (GMAP-2020) and PA$_4$ (yearly measurement; November 12, 2020–October 30, 2021) are
shown in Fig. 3a. These hourly data exhibited a fair logarithmic relationship ($R = 0.45$), and a
relatively lower 1:1 linear relationship ($R = 0.41$), where the linear relationship with PC-NO$_2$
weakened as SI-NO$_2$ levels increased. It appears that the SI-NO$_2$ has a distinct diurnal change
despite the same PC-NO$_2$, and higher variable surface NO$_2$ levels may attribute to relatively
lower linear relationship between PC-NO$_2$ and SI-NO$_2$. To explore these anti-correlation cases
further, we selected the lower and upper bounds of the tendencies; these are plotted in Fig. 3b,
which shows that PC-NO$_2$ was positively correlated with SI-NO$_2$ on February 24, 2021 ($R =$
0.88), while a negative correlation occurred on April 21, 2021 ($R = -0.88$); thus, there was a
wide range of case-specific correlations.
Generally, in remote and clean regions such as the Pacific Ocean, local NO$_2$
concentrations are considered to be at background level, and can be used to represent
stratospheric NO$_2$ amounts. The background level in our study would ideally correspond to the
intercept of the regression model in the PC-SI NO$_2$ scatter diagram (Fig. 3a). In our analysis of
yearly measurements, the intercept of 0.09 DU was consistent with stratospheric NO$_2$ amounts
(0.10 ± 0.02 DU) estimated from the tropospheric monitoring instrument (TROPOMI) at a





nadir pass time of approximately at 1330.

**4.2 Impacts of meteorological conditions on correlations between PC-NO$_2$ and SI-NO$_2$**
Our k-means cluster analysis distinguished three groups with the lowest within-group variance
and largest among-group variance. Among the total of 141 cases, 47, 66, and 28 were classified
into groups 1–3, respectively. Thus, group 2 had the largest proportion of cases (47%) and
group 3 had the smallest (20%). The combination of meteorological components in group 1
indicated the end of a high-pressure system (Psfc > 0, dPsfc/dt < 0), with southerly winds
(NS850 > 0) bringing warmer air (T925 > 0) to the region, leading to stable atmospheric
stratification and weak surface winds (Fig. 4). This group 1 meteorological mode appeared to
result in very weak NO$_2$ ventilation, which produced the highest PC-NO$_2$ and SI-NO$_2$ values.
Group 3 showed the opposite trend, with strong northerly winds bringing colder air into the
region, leading to an unstable atmosphere and stronger surface winds, and ultimately
decreasing PC-NO$_2$ and SI-NO$_2$ to their lowest levels.

SI-NO$_2$ was approximately twice as high in group 1 than group 3, whereas PC-NO$_2$

showed no significant difference (Fig. 4a). We hypothesized that PBLH might also differ
significantly under these micrometeorological conditions; therefore, we further explored daily
maximum PBLH data from Hybrid Single-Particle Lagrangian Integrated Trajectory
(HYSPLIT) Global Forecast System (GFS) simulations for the 141 cases. The mean simulated
PBLH was 942.1 ± 405.3 m for 2020, which was similar to the annual mean daily maximum
PBLH (1,013.6 m) in Osan (Lee et al., 2013). However, the simulated PBLH differed
significantly among the three groups (767.0 ± 304.8, 923.2 ± 335.3, and 1,280.6 ± 501.2 m for
groups 1–3, respectively). The PBLH for group 3 was 1.7-fold higher than that for group 1
(Fig. 4k). We also detected significant differences among the three groups in synoptic
components of the lower troposphere including W850, as well as in local meteorological





parameters such as the sea breeze index, which is calculated as $SBI = U^2/\Delta T$, where U is Wsfc
(Fig. 4c) and $\Delta T$ is the temperature difference between T925 and the sea surface temperature.
Thus, the SBI represents the ratio between inertial ($\rho U^2/2$) and buoyance ($\rho g \beta \Delta T$) forces, where
$\rho$ is air density, $\beta$ is specific heat, and $g$ is gravity, and its value provides an indication of the
likelihood of local circulation events such as sea breezes; at high SBI values, sea breezes cannot
overcome the prevailing wind, whereas low SBI values can indicate strong sea breezes. In the
example shown in Fig. 4l, the SBI values of groups 1–3 were $0.1 \pm 4.5$, $0.1 \pm 9.2$, and $-0.2 \pm$
12.5, respectively. Groups 2 and 3 had similar mean SBI values suggesting little local
circulation; however, group 1 corresponded to dominant local circulation (LD), group 2 to a
mixture of local and synoptic-scale circulation (MD), and group 3 to dominant synoptic-scale
circulation (SD). These results indicate that Seosan may experience frequent LD conditions
(with sun on one third of the days of the year), with infrequent SD conditions (one fifth of all
days).

If the $NO_2$ profiles are vertically uniform within the PBL (e.g., Fig. 4), the mean SI-$NO_2$

under LD and SD conditions can be scaled to PC-$NO_2$ amounts of 0.87 and 0.90 DU at 1 atm
and 298 K, respectively, for the given mean PBLH. The estimated PC-$NO_2$ amounts appeared
to be similar across groups, and yet higher than the PC-$NO_2$ observations (0.31 and 0.32 DU,
respectively), indicating that $NO_2$ profile shapes may deviate slightly from the constant vertical
shape of the PBL.

**4.2.1 Relationship between daily mean PC-$NO_2$ and SI-$NO_2$ under LD, MD, and SD**
**conditions**
Scatter diagrams of daytime mean PC-$NO_2$ and SI-$NO_2$ measurements at Seosan over the entire
1-year period are shown in Fig. 5. Based on the 141 cases, daytime mean values averaged
between 1100 and 1700 KST were used to reduce the effect of nocturnal PBLH variation. Other



data selection criteria included concurrent PC-$NO_2$ and SI-$NO_2$ measurements, with data
acquisition rates of > 80% per day. Overall, PC-$NO_2$ and SI-$NO_2$ were strongly correlated ($R$
= 0.73; Fig. 5), suggesting that the vertical profiles were generally uniform in the PBL
throughout all four seasons. The slope of the linear regression curve shown in Fig. 5a was 0.02
DU/ppb (= $0.53 \times 10^{15}$ molecules cm$^{-2}$/ppb), which is comparable to values ($0.3–0.59 \times 10^{15}$
molecules cm$^{-2}$/ppb) obtained previously in a study of surface and OMI-$NO_2$ measurements
downwind of strong point sources in Israeli cities (Boersma et al., 2009). The intercept (0.17
DU) was within the range of previous Anmyeondo Pandora measurements, suggesting that
intercepts of 0.15–0.2 DU may represent the local background PC-$NO_2$ amount (including the
stratospheric $NO_2$), rather than the influence of local anthropogenic $NO_2$ emissions.

We classified daily averaged PC-$NO_2$ and SI-$NO_2$ data according to the three

meteorological conditions (LD, MD, and SD) and detected a weak correlation under LD
conditions (Fig. 5b); the lowest coefficient of determination for the LD condition ($R^2 = 0.34$)
was approximately half of those for the MD (0.359) and SD (0.64) conditions, suggesting that
$NO_2$ vertical profiles were more complex under LD conditions, with anomalous layers.

**322    4.2.2 Diurnal variations in column-surface $NO_2$ under LD, MD, and SD conditions**

Diurnal patterns of PC-$NO_2$, SI-$NO_2$, and $O_3$ under SD, MD, and LD conditions are shown in
Fig. 6. Under LD conditions, PC-$NO_2$ increased from morning to afternoon (Fig. 6a), whereas
under SD conditions, it had a weak morning peak and subsequent decrease until late afternoon
(Fig. 6c). Under MD conditions, PC-$NO_2$ had one large peak in the morning and a shoulder
peak in the late afternoon (Fig. 6b). However, SI-$NO_2$ showed nearly identical diurnal patterns
among the three meteorological conditions, with an early-morning peak followed by a second
peak in the late afternoon (Fig. 6d–f). Diurnal patterns of $O_3$ were strongly associated with $O_3$-
$NO_2$ photochemical reactions under both LD and MD conditions (Fig. 6g–h), whereas no



particular photochemical effects were detected under SD conditions (Fig. 6i).

A simple linear regression was applied to daytime-average (1100–1700LST)

measurements of both PC-NO$_2$ and SI-NO$_2$ under the three meteorological condition, and
yielded correlation coefficients ($R$) of 0.51 and 0.41 for SD and MD conditions, respectively;
however, LD conditions produced a significantly lower $R$ (0.27). Thus, under SD conditions,
strong synoptic winds suppressed PC-NO$_2$ and SI-NO$_2$ diurnal fluctuations, rendering them
similar to each other. Strong winds also inhibited local effects of O$_3$ formation on the diurnal
variation of PC-NO$_2$, and the smaller impact of chemical conversion from local NO$_2$ to O$_3$
lowered $R$ values during the day. Under MD conditions, both PC-NO$_2$ and SI-NO$_2$ exhibited
distinctive peaks in the morning with a degree of time lag; both subsequently declined toward
noon, and showed higher $R$ values than those obtained under SD conditions. By contrast, under
MD conditions, correlations were enhanced due to a minimum around 1500 KST for both PC-
NO$_2$ and SI-NO$_2$, despite time lags in both peaks in the morning and afternoon.

Previous studies of the Megacity Air Pollution Seoul (MAPS-Seoul) and KORUS-AQ

campaigns reported a typical pattern of continuously increasing PC-NO$_2$ over the Seoul
metropolitan area. However, in the current campaign, we found similar results only under LD
conditions. The diurnal patterns reported in previous studies were mainly caused by the
dominance of NO$_2$ emission sources over NO$_2$ losses (Chong et al., 2019; Herman et al., 2018)
among several processes associated with NO$_2$ photochemical loss, including transport and
deposition, which were also investigated in specific cases in the current study.

**4.3 Aircraft measurements collected during GMAP-2020**
Data collected via aircraft during GMAP-2020 are summarized in Table 2. A total of nine
aircraft measurements were conducted during the campaign period (November 12, 2020–
January 20, 2021). Four of nine flights were conducted under LD conditions, and the remaining



flights (except that on November 27, 2020) were conducted under MD conditions. No aircraft
measurements were consistent with SD conditions during the GMAP-2020 campaign.

We examined spiral segments from each flight over Seosan during 1100–1700 KST to

exclude marginal effects of diurnal variation in $NO_2$ (Fig. 2). The overall results indicated that
the vertical $O_3$ profiles were relatively constant in the PBL, whereas $NO_2$ profiles appeared to
be highly dependent on meteorological conditions. We compared data collected during flights
conducted under LD (one flight) and MD conditions (two flights) during the GMAP-2020
campaign, to examine differences in the vertical structures of the PA and SI observations.

Aircraft measurements of vertical $NO_2$ and $O_3$ profiles for flights FL-5 (December 6) and

FL-6 (December 8) under LD conditions are shown in Fig. 7, along with 24-h backward
trajectories starting at different altitudes (100, 500, and 1,000 m). All observed $NO_2$ profiles
shown in Fig. 7 appeared to have generally exponential curves, with anomalous features at
higher altitudes. For example, when vertical turbulent mixing prevailed within the PBL ($O_3$
profile, Fig. 7b), the data were fitted with an exponential vertical curve, and the anomalous
$NO_2$ layer aloft was found to have a height of 1.5 km, which was higher than the estimated
PBLH of 1.2 km. HYSPLIT 24-h backward trajectories starting at 1200 KST showed that all
air mass from the surface to the lower free atmosphere was transported over the Yellow Sea
via the Shandong Peninsula (Fig. 7c). This finding suggests that the anomalous $NO_2$ layer aloft
was not produced locally (i.e., from local LPS emissions), but instead traveled via long-range
regional-scale transport. According to Anmyeondo Lidar measurements for December 6
(http://kalion.kr), the anomalous $NO_2$ layer aloft corresponded well to an aerosol layer that
appeared at ~1.0 km at approximately 1200 KST, persisting until 2200 KST. However, based
on a cross-comparison of our data, high surface levels of SI-$NO_2$ (> ~4 ppb; Fig. 7a) were
influenced more by local LPS than by that in the atmosphere aloft due to long-range transport
(Fig. 7a).





Aircraft measurements for flight FL-7 (December 9) under LD conditions are shown in
Fig. 7b. The $NO_2$ vertical profile exhibited an exponential curve, with an anomalous peak at
~600 m immediately above the top of the simulated PBL. HYSPLIT backward trajectory data
starting at 1200 KST showed that the non-surface air had a different origin from the surface air
(Fig. 6d), indicating that the anomalous $NO_2$ plume likely traveled from coal-fired power plants
in a nearby industrial city (Taean) northwest of Seosan. This finding indicates a distinct vertical
structure of higher $NO_2$ at the surface due to strong local emissions, whereas lower $NO_2$ levels
were observed at higher altitudes, with anomalously high $NO_2$ levels in some layers aloft due
to medium-range transport from nearby areas. Thus, despite the limited number of aircraft
measurements, the elevated anomalous $NO_2$ structure that was observed intermittently led to a
negative correlation between PA-$NO_2$ and SI-$NO_2$. Therefore, GEMS validation should
proceed cautiously when only surface measurements alone are obtained under LD
meteorological conditions.
Aircraft measurements were conducted under MD conditions on flights FL-1 (November
26), FL-3 (November 28), and FL-8 (December 12) (Fig. 8). We applied several regression
models (linear, exponential, and polynomial) to three vertical structures, and obtained two
distinct $NO_2$ vertical profile patterns from the surface to the PBLH: decreasing linearly for FL-
1 and FL-8 (Fig. 8), and constant with altitude for FL-3 (Fig. 8b). None of the three cases
showed anomalous layers above the PBLH, similar to the exponentially declining profiles
obtained under LD conditions (Fig. 7). These vertical structures observed under MD conditions
may have been induced by strong vertical mixing within the PBL, supplemented by prominent
surface photochemical losses at the same time. The vertical $O_3$ profile during FL-1 showed a
decoupled structure, with different patterns within and above the PBL (Fig. 8d); however, the
other 2 days showed uniform distributions, with no particular anomalous features between the
upper PBL and surface atmosphere (Fig. 8b, c, e, f). The observed daily maximum sensible





heat fluxes measured at Seosan (Fig. S1) were much higher for FL-3 (175.9 $Wm^{-2}$) than FL-1
and FL-8 (118.9 and 102.0 $Wm^{-2}$), suggesting that vertical turbulent mixing was much more
prominent during FL-3. These chemical and physical characteristics are all related to MD
conditions. Thus, the higher coefficient of determination ($R^2$ = 0.64) obtained under MD
conditions (Fig. 5b) has an important bearing on the absence of irregular or anomalous layers
aloft, with little variation regardless of the shape of the curve (Figs. 7 and 8).

**4.4 Analyses of column–surface relationships for specific GMAP-2020 cases**

Figure 9 shows examples of PC-$NO_2$ and SI-$NO_2$ diurnal variation under LD (FL-5 and FL-7)
and MD (FL-1 and FL-8) conditions, and Fig. 10 shows latitudinal mean distributions for FL-
5 and FL-7, based on the aircraft measurement data shown in Figs. 7 and 8. PC-$NO_2$ was found
to be decoupled from SI-$NO_2$ on 2 days, FL-5 and FL-7, which were both classified as having
LD conditions (Fig. 9a, b), whereas good vertical mixing and uniform $NO_2$ distribution were
observed on the remaining 2 days, FL-1 and FL-8, which showed MD conditions (Fig. 9c, d).
According to our analysis of the aircraft measurements (Fig. 7), the poor correlations between
PC-$NO_2$ and SI-$NO_2$ captured by FL-5 and FL-7 were mainly due to an $NO_2$ polluted layer
transported aloft, as described in Section 4.3.

**4.4.1 LD conditions**

Several cases showed poor correlation between PC-$NO_2$ and SI-$NO_2$ under LD conditions
within the study period. When we examined the results of previous studies (Thompson et al.,
2019; Kim et al., 2021; Chong et al., 2019; Herman et al., 2018), we first considered the
possibility that LPS emissions influenced downwind regions under LD conditions, because the
increase in PC-$NO_2$ (but not SI-$NO_2$) may have required an additional source of $NO_2$ apart
from early afternoon traffic emissions. The FL-5 data for December 6 represent an example of





this, showing a poor correlation between PC-NO$_2$ and SI-NO$_2$ ($R^2 = 0.06$; Fig. 9a). On the same
day, Anmyundo LIDAR detected two elevated aerosol layers at 1200 and 1600–2200 KST
(http://kalion.kr); the first aerosol layer may reflect a PC-NO$_2$ peak, as shown in Fig. 9a. The
HYSPLIT backward trajectories, starting at different altitudes from the surface to the lower
troposphere, revealed that all air parcels moved eastward from China to Anmyundo and Seosan
(Figure 1); thus, other NO$_2$ plumes may have begun to pass over Seosan at 1600 KST (Fig. 7c).
Longitudinal SI-NO$_2$ distributions (Fig. 10) exhibited 5.2 ppb at 126.1°E, 8.1 ppb at 126.3°E,
and 7.3 ppb at 126.4°E, averaged between 1300 and 1600 KST by longitude (Table S1),
whereas they were nearly constant at a height of 500–600 m on December 6. Therefore,
westerly winds advected cleaner air from Padori (AQM$_1$) to Seosan at the surface, but not at a
height of 500–600 m, contributing to low SI-NO$_2$ levels in the afternoon (Fig. 9a).

Another example of a weak correlation was obtained by flight FL-7 (December 9), as

shown in Fig. 9b. Time series PC-NO$_2$ data exhibited several peaks during 1200–1400 KST
(Fig. 9b), whereas SI-NO$_2$ showed less temporal variation, resulting in a weak correlation ($R =$
$-0.24$) compared with the overall daytime (1100–1700 KST) correlation ($R^2 = 0.53$; Fig. 5a).
Latitudinal NO$_2$ levels at high altitudes of ~600 m (Fig. 10b) gradually increased northward,
whereas surface NO$_2$ was minimal at the midpoint. For example, at high altitudes, the
latitudinal mean NO$_2$ levels were 1.4 ppb (36.8°N), 4.1 ppb (36.9°N), and 5.1 ppb (37.0°N),
whereas the SI-NO$_2$ levels at the same sites were 18.0 ppb (36.8°N), 14.3 ppb (36.9°N), and
16.8 ppb (37.0°N), respectively, averaged during 1200–1400 KST by latitude (Table S1). This
finding is attributable to a prevailing north wind that transported NO$_2$ southward at high
altitudes, while simultaneously ventilating SI-NO$_2$ toward outer Seosan, resulting in the
development of several PC-NO$_2$ peaks. In contrast, SI-NO$_2$ decreased slowly (Figs. 9b and
10b).



### 4.4.2 MD and SD conditions

We obtained higher PC–SI correlation coefficients under MD and SD conditions than LD conditions (Figs. 5b and 9c, d). Under MD and SD conditions, diurnal variation in PC-NO$_2$ and SI-NO$_2$ showed simultaneous declines from early morning until noon (Fig. 6). Notably, PC-NO$_2$ showed a continuously decreasing trend, particularly during the morning hours, in the period of approximately 0900–1200 KST under both MD and SD conditions (Fig. 6b, c). These diurnal patterns of decreasing PC-NO$_2$ in the study area were opposite to those reported in previous studies (Chong et al., 2019; Herman et al., 2018) that observed increasing PC-NO$_2$ in large urban areas during thr daytime, caused by higher NO$_2$ emissions even during photochemical NO$_2$ losses to form O$_3$.

In this study, we hypothesized that decreasing PC-NO$_2$ can occur due to photochemical loss and surface wind transport, which both intensify with increasing solar radiation in the morning. Photochemically, NO$_2$ is converted into photochemical oxidants such as PAN, HNO$_3$, and nitrate under sunlight, thereby disrupting the NOx–VOC–O$_3$ cycle. Concurrently, Wsfc intensified due to thermal turbulence transport of NO$_2$ emissions away from Seosan during the day. Thus, PC-NO$_2$ decreases under MD conditions as a result of ventilation effects caused by stronger wind speeds. There are two possible mechanisms for this: sea breeze penetration (because the study area is adjacent to the northern coast of the Taean Peninsula; Fig. 1) and vigorous turbulent mixing (which leads to vertical mixing of surface NO$_2$ during PBL growth; Sun et al., 2013). We investigated these factors in detail for specific cases.

Figure 11 shows the diurnal variation in selected meteorological and chemical variables measured under MD (November 25) and SD conditions (December 14). Under MD conditions (Fig. 11a–c), declines in PC-NO$_2$ and SI-NO$_2$ were observed toward noon. In particular, decreasing PC-NO$_2$ was accompanied by increased Wsfc (Fig. 11b); therefore, we examined GMAP-2020 campaign measurements of sea breeze penetration.



Figure S2a shows diurnal variation in observed air temperatures at site $Met_1$ and
measured sea surface temperatures at nearby site $Met_2$ (37.14°N, 126.01°E), located 55 km
from $PA_4$. The thermal meteorological observations were used to calculate SBI (+0.37), which
was greater than +3 (the threshold for sea breeze occurrence; Brigges and Graves, 1962). Sea
breeze disturbances with a sharp decrease (increase) in temperature (humidity) were observed
at site $Met_3$ (Fig. S2b), which is located on the northern coastline of Taean Peninsula (Fig. 1).
However, sea breezes did not progress inland at the $Met_1$ Seosan Meteorological Automated
Surface Observing System (ASOS) site, which is closer to the Pandora sites; sea breezes did
not correlate with $NO_2$ ventilation to offset its high emission.
We further detected a strong positive correlation between wind speed and sensible heat
flux (Fig. 11b). We speculated that thermal and momentum turbulences caused by a vertical
temperature gradient and surface friction entrained surface turbulence, thus increasing
momentum in the free atmosphere downward to the surface due to strong turbulent mixing
within the PBL, in turn leading to a uniform vertical $NO_2$ profile with a positive correlation
between $PC\text{-}NO_2$ and $SI\text{-}NO_2$. Figure S3 shows a comparison of daily maximum sensible heat
and momentum fluxes under LD, MD, and SD conditions during the GMAP-2020 campaign.
SD conditions showed the highest mean heat flux, followed by MD and LD, indicating that
downward momentum transport led by both heat and momentum fluxes plays a greater role in
$W_{sfc}$ enhancement under MD than LD conditions within the PBL.
Photolytic $NO_2$ loss was detected as temporal variations in $NO_2$, $NO_3^-$, and CO at $PA_4$.
Because no $NO_2$ analyzer was installed at $PA_4$, $NO_2^*(= NO_y - NO)$ was used instead of $NO_2$
under the assumption that $NO_z$ is negligible in winter. Figure 11c shows the diurnal variation
in $NO_2$, $O_3$, and $NO_3^-$ under MD conditions, normalized by CO to reduce the effect of PBL
evolution. The results showed that $NO_2/CO$ decreased after the morning peak; however, $NO_3^-$
/CO and $O_3/CO$ increased toward midday, indicating that photolytic activity also contributed


considerably to the concurrent decline of SI-NO$_2$ and PC-NO$_2$ (Fig. 11a). In turn, this indicated
that photochemistry can contribute to higher correlation coefficients under MD conditions.
Under SD conditions (Fig. 11d–f), PA-NO$_2$ and SI-NO$_2$ exhibited weak diurnal
variability compared with LD and MD conditions. SD conditions on December 14 produced
significantly stronger winds (i.e., wind speed > 6 m s$^{-1}$ at 1300 KST), with generally higher
PBLHs (Fig. 11e). Meteorological features, such as strong wind at both 850 hPa (18.0 m s$^{-1}$)
and 10 m height (4.26 m s$^{-1}$), suppressed both PC-NO$_2$ and SI-NO$_2$ (7.3 ppb and 0.31 DU,
respectively) to below the average, producing a strong correlation ($R = 0.9$ at AQM$_5$) and nearly
flattening their temporal curves during the day (Fig. 11d). Thus, under SD conditions, wind
speed and turbulent fluxes such as sensible heat flux had larger values, and NO$_2$ and NO$_3^-$
decreased or increased at the same time during the day (Fig. 11f), indicating that the transport
effect was much greater than that of local photochemical loss over the study area.
In conclusion, in this case-specific study, we discussed correlations between PC-NO$_2$ and
SI-NO$_2$, and explored their mechanisms by investigating the impact of meteorological and
photochemical conditions. A weak correlation between PC-NO$_2$ and SI-NO$_2$ occurred when
anomalously high concentrations remained, with ragged fragments of NO$_2$ plumes in the upper
or middle layers. We also found that a negative correlation occurred intermittently under LD
conditions, with generally lower PBLH. In particular, elevated pollutant levels due to regional-
scale transport or decoupled NO$_2$ plumes advected within the PBL may also have caused the
weak correlation between PC-NO$_2$ vs. SI-NO$_2$. These phenomena were detected only from the
PA–SI coupled measurements in this study. Thus, when either PC or SI observations are
applied alone for GEMS validation, undetected bias can occur under LD conditions,
particularly where transport processes prevail.

**5. Conclusions**





In this study, we explored the potential applicability of combined PC-NO$_2$ and SI-NO$_2$
measurements collected at Seosan during the GMAP-2020 campaign. We characterized the
correlation between PC-NO$_2$ and SI-NO$_2$ under various conditions to understand the complex
air quality of Seosan, which appears to be vulnerable to LPS emissions from surrounding areas.
We hypothesized that correlations between PC-NO$_2$ and SI-NO$_2$ are closely related to NO$_2$
vertical profiles, which also depend on meteorological conditions. We performed statistical
analyses of a year-long PC-NO$_2$ dataset (November 12, 2020–October 30, 2021) combined
with meteorological data, *in situ* ground data, and airborne chemical data measured during the
GMAP-2020 campaign in the same period.

Our results showed that hourly PC-NO$_2$ and SI-NO$_2$ over the 1-year period exhibited a

logarithmic relationship with a fair correlation ($R = 0.45$), and the intercept of the logarithm
regression line (corresponding to zero-surface NO$_2$) was 0.09 DU, consistent with the
stratospheric column NO$_2$ amounts retrieved by TROPOMI. Daily mean PC-NO$_2$ and SI-NO$_2$
exhibited a good linear correlation ($R = 0.73$), supporting the overall uniformity of NO$_2$ profiles
in the PBL over Seosan despite the continuous impact of LPS emissions.

The impact of meteorological conditions on the relationship between PC-NO$_2$ and SI-

NO$_2$ was investigated through agglomerative hierarchical clustering, which indicated three
meteorological conditions: LD, MD, and SD. Under LD conditions, southerly winds advect
warm air under the upper ridge, forming stable and short PBLs and weak surface winds. By
contrast, under SD conditions, cold northerly winds induce unstable and high PBLs with strong
surface winds. The correlations between daily mean PC-NO$_2$ and SI-NO$_2$ levels, and their
variations during 1100–1700 KST, weakened under LD conditions, suggesting that the shape
of the NO$_2$ profile typically deviates from a uniform profile under SD and MD conditions.
Aircraft measurements under LD conditions demonstrated NO$_2$ plumes aloft, with anomalous
vertical structures and different horizontal (latitudinal) gradients of surface NO$_2$ at higher





altitudes, such as 600 m over Seosan.

Thus, the relationship between PC-$NO_2$ and SI-$NO_2$ depends on the presence of $NO_2$

plumes aloft under LD conditions, which provide a favorable environment for LPS plumes
decoupled from the surface at Seosan. The findings of this study suggest that the correlation of
PC-$NO_2$ and SI-$NO_2$ may serve as an indicator of the degree of complexity of urban air quality.
This correlation can be optimally applied for air quality evaluation and environmental satellite
validation by combining the Pandora Asia Network with AQM networks. More detailed studies
on urban air pollution evaluation will be undertaken based on PC, DOAS, aircraft, SI air
quality, and surface turbulence observation data, as well as modeling studies of data collected
during the GMAP-2021 campaign.

**Acknowledgments**
We thank all those who contributed to the GMAP-2020 field campaign, and PGN for raw data
processing.

**Funding**
This study was supported by the National Institute of Environmental Research (NIER-2021-01-01-052
and NIER-2021-03-03-001), and was partially supported by National Research Foundation of Korea
(NRF) funded by the Ministry of Education of the Republic of Korea (Grant No.
2020R1A6A1A03044834)

**Data Availability**

**Author Contributions:**





Lim-Seok Chang: Conceptualization, Formal analysis, Visualization, Investigation, Writing -
Original draft; Donghee Kim, Hyunkee Hong, Deok-Rae Kim, Jeonga Yu, and Daewon Kim; Data
curation; Hanlim Lee, Kwangyul Lee,  and Jinkyu Hong: Methodology and  formal analysis; Hyun-
Young Jo: Formal analysis and Visualization; Cheol-Hee Kim ; Writing—original draft preparation,
Writing—review and editing. All authors have read and agreed to the published version of the
manuscript

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



**List of Tables**







**Table 1.** Summary of NO$_2$ column data from four Pandora (PA) measurement sites.

| Site | Site name | Site location | | Mean (DU) | SD (DU) | Minimum (DU) | Maximum (DU) | Number of data points (days) | Operating period |
|------|-----------|-----------|---------|-----------|---------|--------------|--------------|------------------------------|------------------|
| | | Longitude (°E) | Latitude (°N) | | | | | | |
| PA$_1$ | Seosan-DHJ | 126.502 | 36.900 | 0.50 | 0.22 | 0.20 | 1.60 | 838 (11) | GMAP-2020 campaign |
| PA$_2$ | Seosan-DM | 126.458 | 36.778 | 0.43 | 0.19 | 0.18 | 1.62 | 1241 (13) | GMAP-2020 campaign |
| PA$_3$ | Seosan-CC | 126.449 | 36.785 | 0.40 | 0.14 | 0.18 | 0.97 | 1242 (13) | GMAP-2020 campaign |
| PA$_4$ | Seosan-SS | 127.492 | 36.777 | 0.39 | 0.16 | 0.17 | 1.79 | 8753 (141)* | 1 year (Nov. 12, 2020– Oct. 30, 2021) |

* The Pandonia Global Network (PGN) retrieval algorithm was applied to yearly
measurements.







**Table 2.** Summary of aircraft measurements collected during the Geostationary Environment
Monitoring Spectrometer (GEMS) Map of Air Pollution (GMAP)-2020 campaign period
(November 12, 2020–January 20, 2021).

| Flight no. | Date | Meteorological classification |
|---|---|---|
| FL-1 | Nov. 26, 2020 | MD [1] |
| FL-2 | Nov. 27, 2020 | No Pandora measurements |
| FL-3 | Nov. 28, 2020 | MD |
| FL-4 | Dec. 1, 2020 | LD [2] |
| FL-5 | Dec. 6, 2020 | LD |
| FL-6 | Dec. 8, 2020 | LD |
| FL-7 | Dec. 9, 2020 | LD |
| FL-8 | Dec. 12, 2020 (am) | MD |
| FL-9 | Dec. 12, 2020 (pm) | MD |

[1] LD: local wind-dominant conditions; [2] MD: mixed conditions.






**Figure Captions**


**Figure 1**. Map of sites used for Geostationary Environment Monitoring Spectrometer (GEMS)
Map of Air Pollution (GMAP) campaigns conducted in (left) Seosan, South Korea in
November 2020 to January 2021 (GMAP-2020), and (right) the Seoul metropolitan area from
October 2021 to November 2021 (GMAP-2021). (Left) Measurement sites around Seosan, the
study area for the GMAP-2020 campaign. Red circles indicate Pandora column measurement
sites including (left) Seosan Daehoji ($PA_1$), Seosan Dongmun ($PA_2$), Seosan City Council
($PA_3$), and Seosan Super Site ($PA_4$). Blue triangles indicate large point sources (LPSs)
including the Taean and Dangjin thermal power stations ($LPS_1$ and $LPS_2$, respectively),
Hyundai steelworks ($LPS_3$), and Daesan petrochemical complex ($LPS_4$). Yellow squares
indicate Automated Synoptic Observing System meteorological sites in Seosan ($Met_1$), AWS
($Met_2$), and buoy ($Met_3$). Green squares indicate air quality monitoring (AQM) network
stations including Padori ($AQM_1$), Leewon ($AQM_2$), Taean ($AQM_3$), Daesan ($AQM_4$),
Seongyeon ($AQM_5$), and Dongmoon ($AQM_6$). In the right panel, the black line indicates the
route used for car-based differential optical absorption spectroscopy (Car-DOAS)
measurements and the blue dotted line indicates the horizontal domain of Geo-CAPE airborne
simulator (GCAS) measurements taken during the GMAP-2021 campaign.

**Figure 2.** Flight tracks for two Cessna Grand Caravan 208 B aircraft over Pandora sites (left)
$PA_4$ and (right) $PA_1$ during the GMAP-2020 campaign. Colored circles indicate airborne $NO_2$
concentration observations. Stacked circles indicate spiral flights conducted over two sites.

**Figure 3.** a) Pandora column (PC) $NO_2$ measurements as a function of surface *in situ* (SI) $NO_2$
observations at Pandora sites $PA_1$–$PA_3$ during the GMAP-2020 campaign and $PA_4$ during a 1-
year period. A logarithmic regression model was used to evaluate the relationship between PC
and SI measurements (black line). (b) Sample scatter plots of PC-$NO_2$ and SI-$NO_2$ for February
24 (red) and April 21 (blue), 2021.

**Figure 4.** K-means clustering yielded three groups of cases for (a) surface $NO_2$ and (b) PC-
$NO_2$, associated with eight meteorological variables: (c) surface wind speed (Wsfc), (d) Psfc,
(e) Psfc tendency (dPsfc/dt), (f) 925-hPa air temperature (T925), (g) 850-hPa wind speed





(W850), (h) 850-hPa north–south wind component (NS850), (i) 850-hPa east–west wind
component (EW850), and (j) 500-hPa geopotential height (GPH500). All data were de-
seasonalized using the 30-day moving average, except PC-$NO_2$, for which the monthly average
was used. (k) Simulated daily maximum mixing height (not directly clustered). (l) Box and
whisker plots of the sea breeze index (SBI) at Seosan for the 1-year period. Red dots indicate
critical SBI values (3: Biggs and Graves, 1962).

**Figure 5.** Scatter plots of daytime mean PC-$NO_2$ vs. SI-$NO_2$ measurements at site $PA_4$ under
(a) all meteorological conditions and (b) each meteorological condition over a period of 1 year
(November 12, 2020–October 30, 2021).
**Figure 6.** Box and whisker plots of diurnal variations in (a–c) PC-$NO_2$, (d–f) SI-$NO_2$, and (g–
i) surface $O_3$ under synoptic wind-dominant (SD), mixed (MD), and local wind-dominant (LD)
conditions in Seosan during a 1-year period (November 12, 2020–October 30, 2021).
**Figure 7.** Box and whisker plots of the vertical $NO_2$ and $O_3$ profiles measured by GMAP
aircraft superposed with *in situ* $AQMS_1$ measurements during flights (a, b) FL-1 (November
26) and (d, e) FL-8 (December 12). Blue dashed lines are linear regression lines fitted to $NO_2$
and $O_3$ profiles within the planetary boundary layer (PBL). Black arrows indicate the simulated
PBL height (PBLH) obtained from the Korea Meteorological Administration (KMA).
HSYPLIT 24-h backward trajectories in Seosan are shown at altitudes of 100, 500, and 1,000
m, starting at 1600 KST on November 26 and 1200 KST on December 12.
**Figure 8.** Box and whisker plots of vertical profiles obtained from GMAP aircraft superposed
with *in situ* AQMS measurements for (1) $NO_2$ and (2) $O_3$ for flights (a) FL-1 (November 26),
(b) FL-3 (November 28), and (c) FL-8 (December 12). Blue dashed lines are linear regression
lines fitted to $NO_2$ and $O_3$ in the PBL. Black arrows indicate PBLH simulated by the Hybrid
Single-Particle Lagrangian Integrated Trajectory (HYSPLIT) Global Forecast System (GFS).
**Figure 9.** Time series and scatter plots of PC-$NO_2$ and SI-$NO_2$ at $PA_2$ on (a) December 6, (b)
December 9, (c) November 26, and (d) December 12. (e) Scatter plot of PC-$NO_2$ and SI-$NO_2$
on December 6 (blue), December 9 (red), November 26 (gray), and December 12 (black). (f)
Vertical potential temperature profiles on December 6, 9, and 12, 2020. Radiosonde data for
November 26, 2020 are missing.


**Figure 10.** Latitudinal NO$_2$ distribution at the surface and 600 m over PA$_4$ (Seosan Super Site),
averaged during (a) 1300–1600 KST on December 6 (FL-5) by longitude and (b) 1200–1400
KST on December 9 (FL-7) by latitude, obtained from airborne (blue) and surface
measurements (red).

**Figure 11.** Example of diurnal variations on November 25 (a, c) and December 14 (d, f). (a, d)
Column NO$_2$ at sites PA$_1$–PA$_4$ and surface NO$_2$ at the air quality monitoring sites AQM$_4$ and
AQM$_6$. (b, e) Sensible heat fluxes and surface wind speed at PA$_4$. (c, f) Diurnal variations in
NO$_2$, NO$_2^-$, and O$_3$ normalized by CO. A map of the measurement sites is shown in Figure 1.




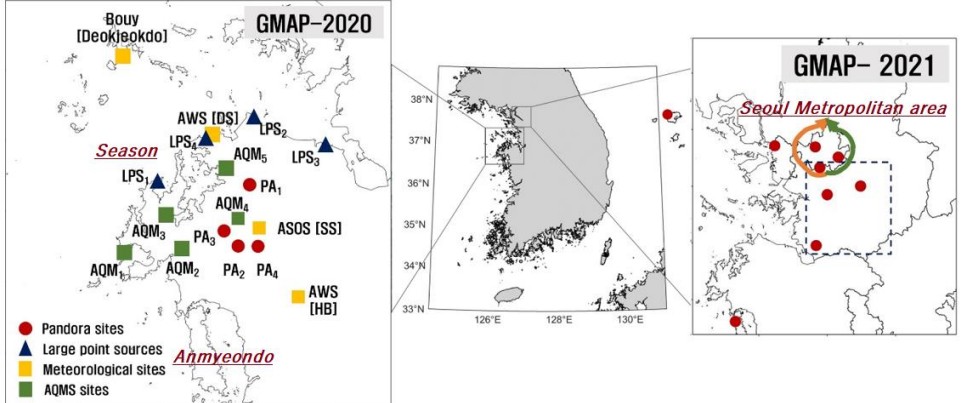



**Figure 1.** Map of sites used for Geostationary Environment Monitoring Spectrometer (GEMS)
Map of Air Pollution (GMAP) campaigns conducted in (left) Seosan, South Korea in
November 2020 to January 2021 (GMAP-2020), and (right) the Seoul metropolitan area from
October 2021 to November 2021 (GMAP-2021). (Left) Measurement sites around Seosan, the
study area for the GMAP-2020 campaign. Red circles indicate Pandora column measurement
sites including (left) Seosan Daehoji ($PA_1$), Seosan Dongmun ($PA_2$), Seosan City Council
($PA_3$), and Seosan Super Site ($PA_4$). Blue triangles indicate large point sources (LPSs)
including the Taean and Dangjin thermal power stations ($LPS_1$ and $LPS_2$, respectively),
Hyundai steelworks ($LPS_3$), and Daesan petrochemical complex ($LPS_4$). Yellow squares
indicate Automated Synoptic Observing System meteorological sites in Seosan ($Met_1$), AWS
($Met_2$), and buoy ($Met_3$). Green squares indicate air quality monitoring (AQM) network
stations including Padori ($AQM_1$), Leewon ($AQM_2$), Taean ($AQM_3$), Daesan ($AQM_4$),
Seongyeon ($AQM_5$), and Dongmoon ($AQM_6$). In the right panel, the black line indicates the
route used for car-based differential optical absorption spectroscopy (Car-DOAS)
measurements and the blue dotted line indicates the horizontal domain of Geo-CAPE airborne
simulator (GCAS) measurements taken during the GMAP-2021 campaign.






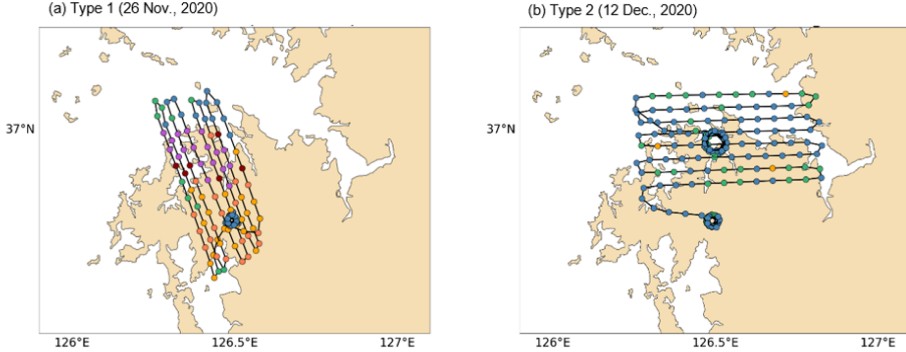


**Figure 2.** Flight tracks for two Cessna Grand Caravan 208 B aircraft over Pandora sites (left)
PA$_4$ and (right) PA$_1$ during the GMAP-2020 campaign. Colored circles indicate airborne NO$_2$
concentration observations. Stacked circles indicate spiral flights conducted over two sites.




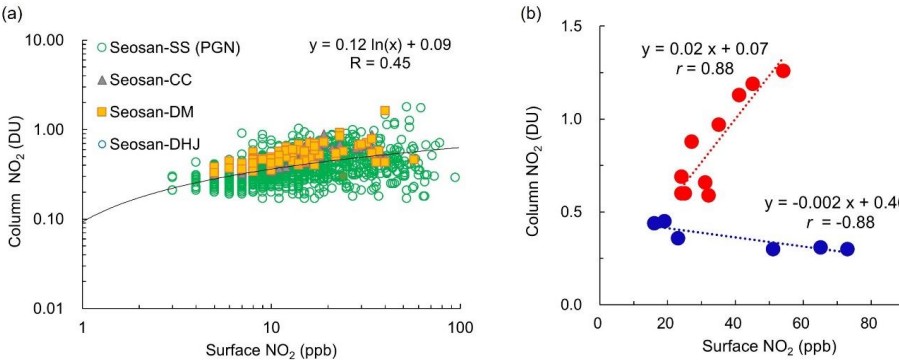


**Figure 3.** a) Pandora column (PC) NO$_2$ measurements as a function of surface *in situ* (SI) NO$_2$
observations at Pandora sites PA$_1$–PA$_3$ during the GMAP-2020 campaign and PA$_4$ during a 1-
year period. A logarithmic regression model was used to evaluate the relationship between PC
and SI measurements (black line). (b) Sample scatter plots of PC-NO$_2$ and SI-NO$_2$ for February
24 (red) and April 21 (blue), 2021.



832

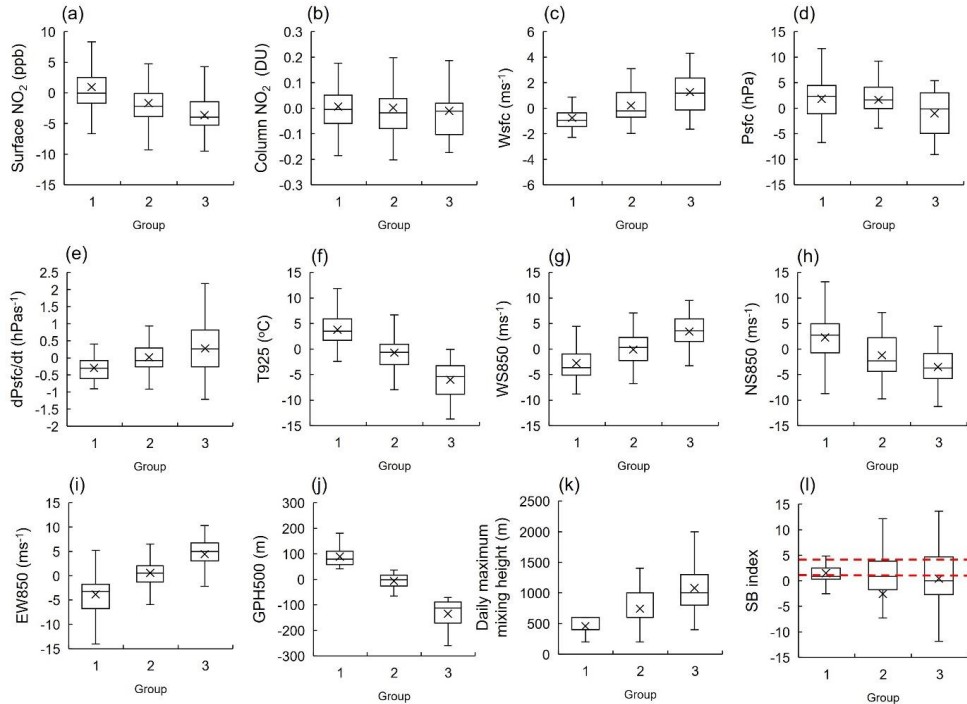

**Figure 4.** K-means clustering yielded three groups of cases for (a) surface $NO_2$ and (b) PC-$NO_2$, associated with eight meteorological variables: (c) surface wind speed (Wsfc), (d) Psfc, (e) Psfc tendency (dPsfc/dt), (f) 925-hPa air temperature (T925), (g) 850-hPa wind speed (W850), (h) 850-hPa north–south wind component (NS850), (i) 850-hPa east–west wind component (EW850), and (j) 500-hPa geopotential height (GPH500). All data were de-seasonalized using the 30-day moving average, except PC-$NO_2$, for which the monthly average was used. (k) Simulated daily maximum mixing height (not directly clustered). (l) Box and whisker plots of the sea breeze index (SBI) at Seosan for the 1-year period. Red dots indicate critical SBI values (3: Biggs and Graves, 1962).










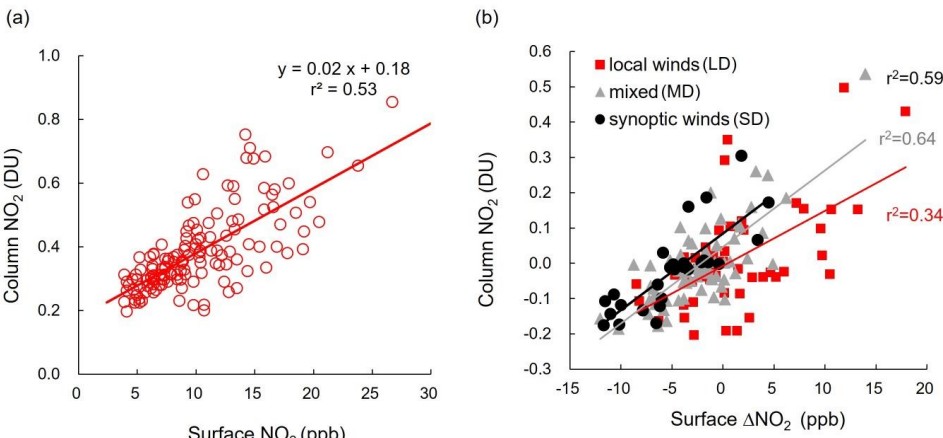

**Figure 5.** Scatter plots of daytime mean PC-$NO_2$ vs. SI-$NO_2$ measurements at site $PA_4$ under (a) all meteorological conditions and (b) each meteorological condition over a period of 1 year (November 12, 2020–October 30, 2021).

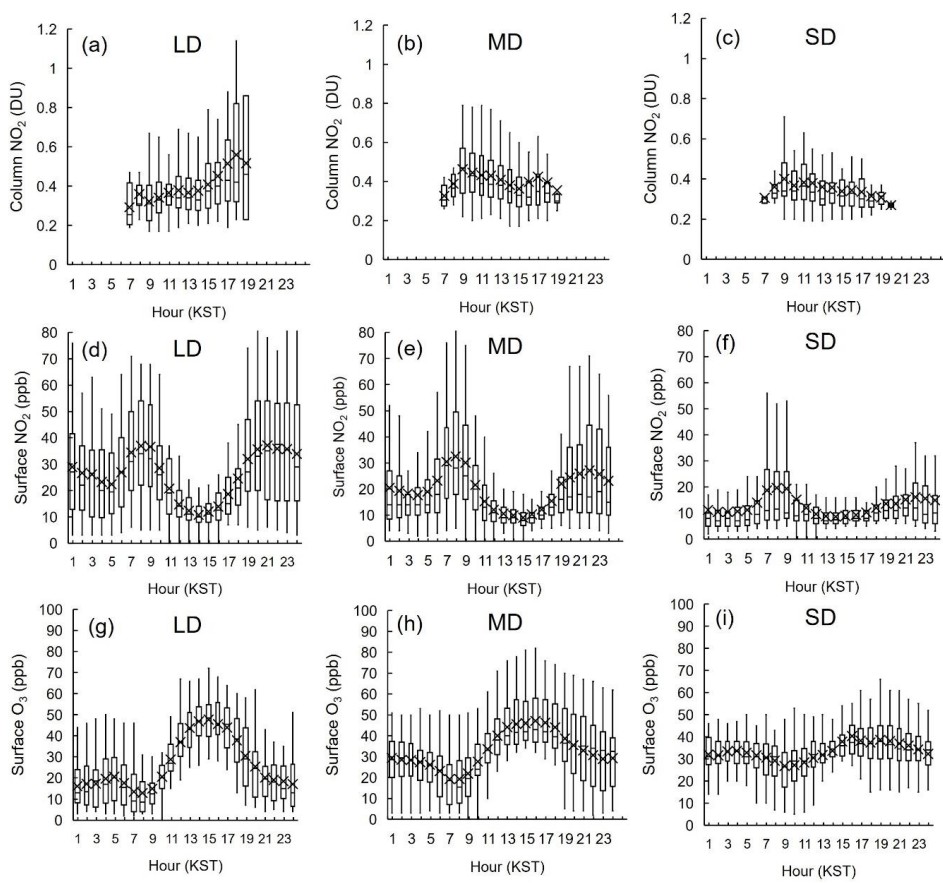

**Figure 6.** Box and whisker plots of diurnal variations in (a–c) PC-NO$_2$, (d–f) SI-NO$_2$, and (g–i) surface O$_3$ under synoptic wind-dominant (SD), mixed (MD), and local wind-dominant (LD) conditions in Seosan during a 1-year period (November 12, 2020–October 30, 2021).




881

882

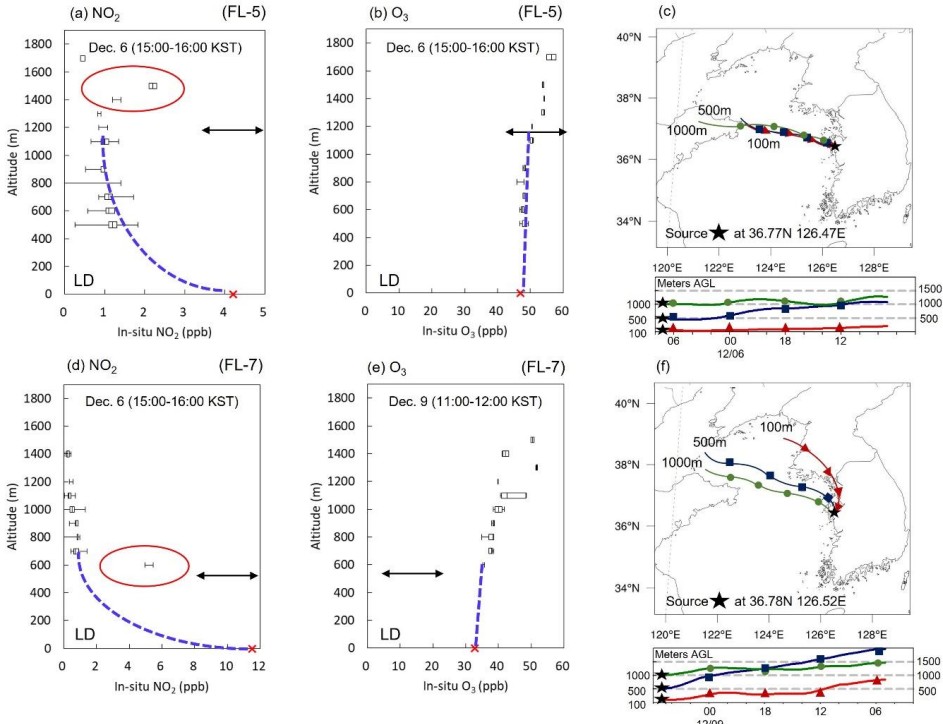

883

**Figure 7.** Box and whisker plots of the vertical NO$_2$ and O$_3$ profiles measured by GMAP aircraft superposed with *in situ* AQMS$_1$ measurements during flights (a, b) FL-1 (November 26) and (d, e) FL-8 (December 12). Blue dashed lines are linear regression lines fitted to NO$_2$ and O$_3$ profiles within the planetary boundary layer (PBL). Black arrows indicate the simulated PBL height (PBLH) obtained from the Korea Meteorological Administration (KMA). HSYPLIT 24-h backward trajectories in Seosan are shown at altitudes of 100, 500, and 1,000 m, starting at 1600 KST on November 26 and 1200 KST on December 12.








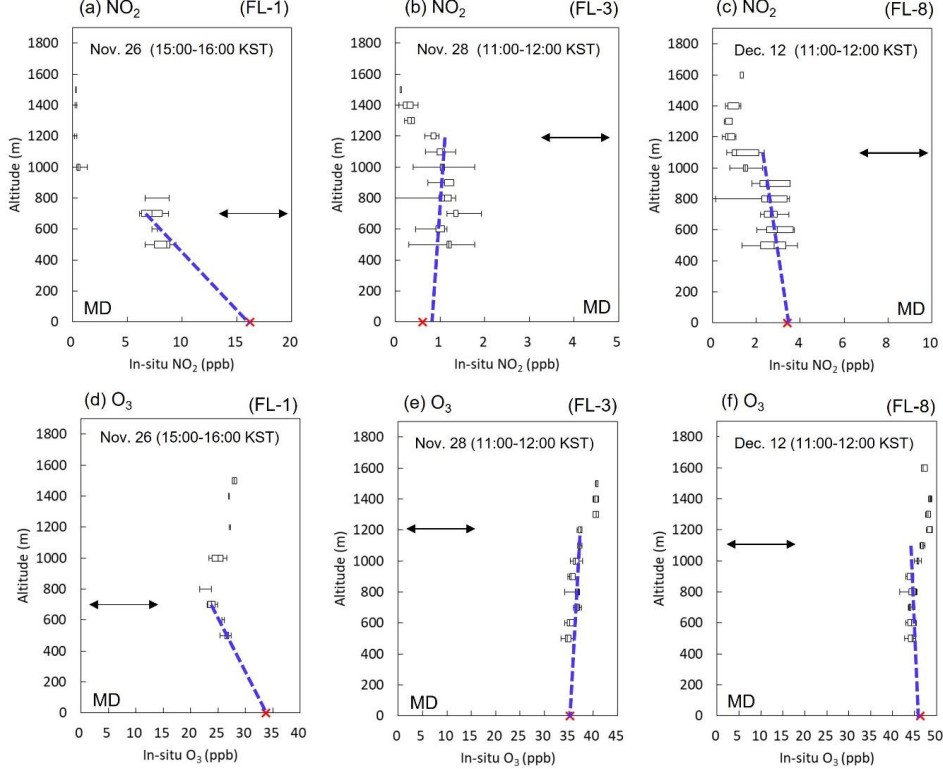


**Figure 8.** Box and whisker plots of vertical profiles obtained from GMAP aircraft superposed
with *in situ* AQMS measurements for (1) $NO_2$ and (2) $O_3$ for flights (a) FL-1 (November 26),
(b) FL-3 (November 28), and (c) FL-8 (December 12). Blue dashed lines are linear regression
lines fitted to $NO_2$ and $O_3$ in the PBL. Black arrows indicate PBLH simulated by the Hybrid
Single-Particle Lagrangian Integrated Trajectory (HYSPLIT) Global Forecast System (GFS).






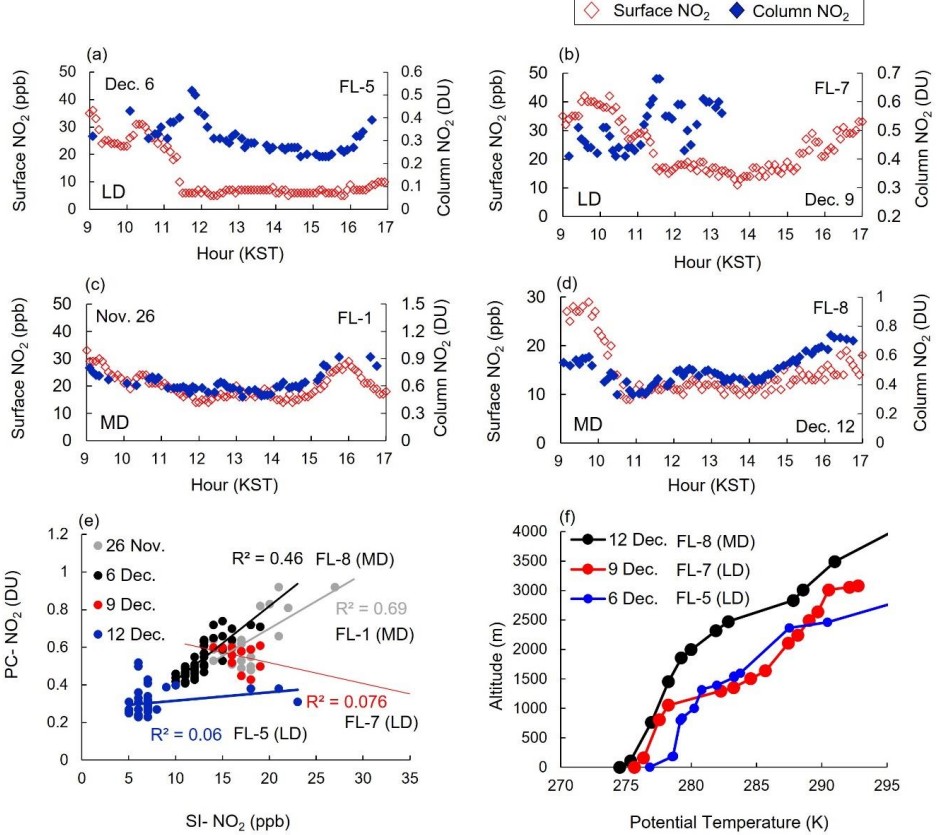


**Figure 9.** Time series and scatter plots of PC-NO$_2$ and SI-NO$_2$ at PA$_2$ on (a) December 6, (b) December 9, (c) November 26, and (d) December 12. (e) Scatter plot of PC-NO$_2$ and SI-NO$_2$ on December 6 (blue), December 9 (red), November 26 (gray), and December 12 (black). (f) Vertical potential temperature profiles on December 6, 9, and 12, 2020. Radiosonde data for November 26, 2020 are missing.
















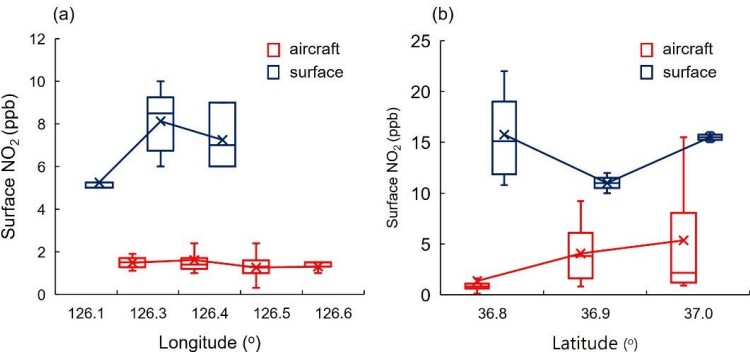


**Figure 10.** Latitudinal NO$_2$ distribution at the surface and 600 m over PA$_4$ (Seosan Super Site),
averaged during (a) 1300–1600 KST on December 6 (FL-5) by longitude and (b) 1200–1400
KST on December 9 (FL-7) by latitude, obtained from airborne (blue) and surface
measurements (red).

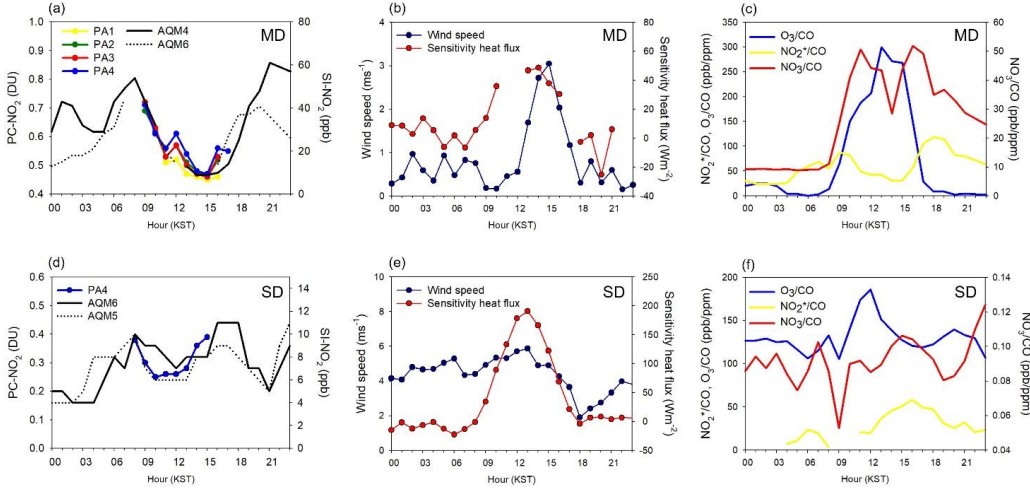


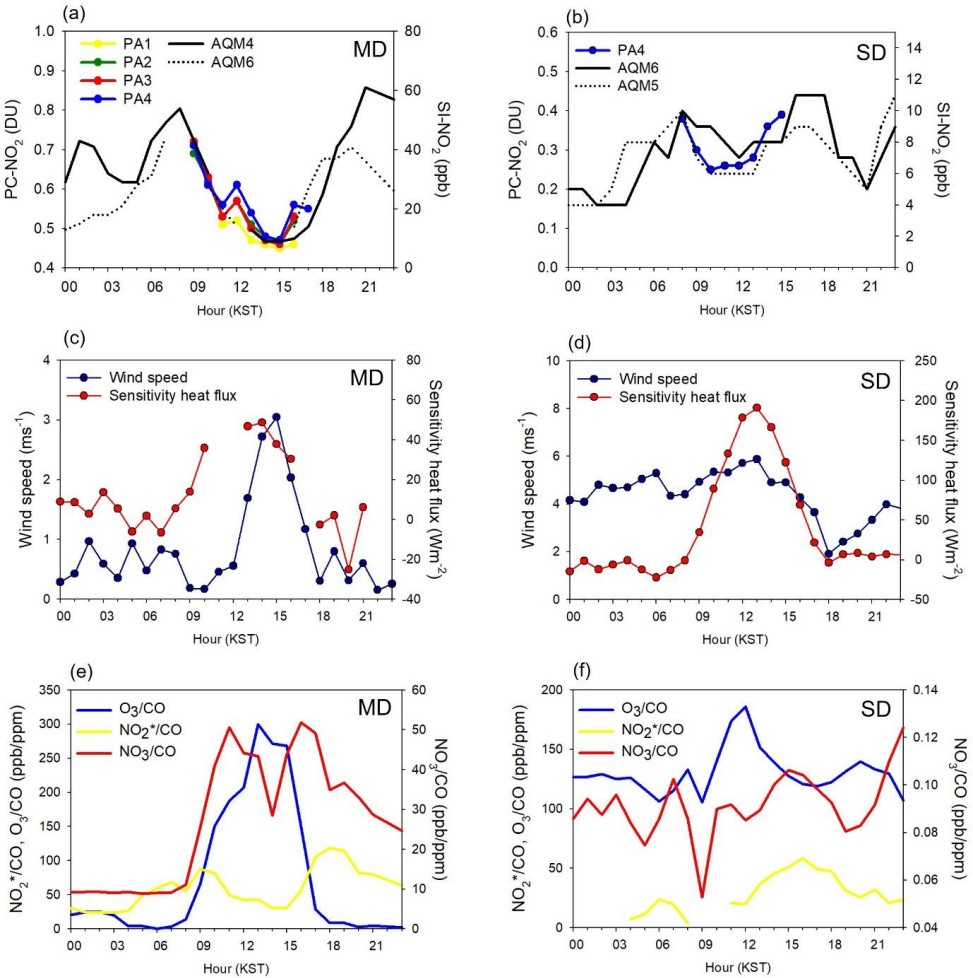


**Figure 11.** Example of diurnal variations on November 25 (a, c) and December 14 (d, f). (a, d)
Column $NO_2$ at sites $PA_1$–$PA_4$ and surface $NO_2$ at the air quality monitoring sites $AQM_4$ and
$AQM_6$. (b, e) Sensible heat fluxes and surface wind speed at $PA_4$. (c, f) Diurnal variations in
$NO_2$, $NO_2^-$, and $O_3$ normalized by CO. A map of the measurement sites is shown in Figure 1.

