# Peer review of "Evaluation of correlated Pandora column $NO_2$ and in situ surface $NO_2$ measurements during GMAP campaign"

_Atmospheric Chemistry and Physics, 2022_

## Author Comment (AC1)

**Answer to Referee # 1**

(Reviewer's comments in BLACK, authors' reply in BLUE, corresponding modifications in the MS (manuscript) in red)

**General Comments:**

This study was performed based on the intensive field measurement data during the GMAP campaign in South Korea. The vision of this campaign is clear, the measurement looks well performed, and the analysis result looks interesting. But the key message is not well transferred to authors; In other words, authors need to spend more efferts to make a main root of this manuscript. This study actually focused on the relationship between the column density and surface level of not associated with the validation of satellite data (GEMS). But authors would like to underline the 'validation' purpose of GMAP, again, which is not clearly related to the main finding of this study. This mismatching makes the whole manuscript very distracted, hurts the organization, and finally results in the poor delivery of key points of this study. Thus, the major revision is largely required.

→ GMAP campaign has been actually started for multi-purpose, including GEMS validation. During GMAP-2021, multi-perspective observations were further obtained from the ground and space; participating remote sensing instruments were MAXDOAS, Car-DOAS, GCAS, and Pandora data. Regarding the revised MS, we limited our interpretation to 2020 campaign only by removing the explanation of GMAP-2021, following the reviewer's point. We appreciate the insightful and useful comments on ways to strengthen our MS. We modified or took out, following the reviewer's suggestion, as bellow.

**Specific Comments:**

**Whole manuscript**: It seems that all results in this manuscript are related the GMAP 2020 (focus on Seosan), not GMAP 2021 (related to the Seoul Metropolitan). I strongly recommend to remove the statement of GMAP 2021 here. Namely, the result of GMAP 2020 is OK enough for this manuscript. It actually makes the key point more obvious because the data analysis using GMAP 2021 was not performed here. This problem is coming from authors' emphasis on the 'validation purpose' of

GMAP campaign. Again, the results in this manuscript are not much related to the validation of GEMS data. This work mainly discussed the similarity/difference between the column and surface NO2. This topic is solely interesting enough. If the 'validation'-related wording is frequently raised in the manuscript, however, the merit of this study (analysis for the relationship between column and surface NO2) becomes weaker and readers would like to see the 'validation result' that is not included in this manuscript. → We agree with reviewer's point; thus, removed GMAP-2021 in the Fig. 1, as well as paragraph described in the MS (Line 181-182, Line 562-565, Line 710-711, 720-723, and etc, in the original MS). Thank you for pointing to this.

**Title**: in this context, I would recommend to change the title, only focusing on the relationship between column and in situ NO2 during the GMAP 2020 campaign. → In the title, observation has been replaced by specific species 'NO$_2$'. Thus, we changed the title to "Evaluation of correlated Pandora column NO$_2$ and in situ surface NO$_2$ measurements during GMAP campaign"

**Line 6 and whole manuscript**: SI implies the 'surface in situ', but here the in-situ NO2 obtained from the aircraft measurement is rather utilized for the analysis (e.g., Figs. 7 and 8), which was collected in the same GMAP 2020. → We also agree on this reviewer's point. In our text, however, there has been a need to distinguish surface 'in situ' and in-situ aircraft measurement: thus, we concurred that aircraft measurement (without mentioning 'in-situ') would be concise to deliver our results to Journal reader.

**Line 9-10**: This is a repetition of previous statements in line 5-6. → Done. We removed the redundant phrase. In the revised text, Line 9-10 were now replaced by 'we explored the synergy of combined analysis of both PC-NO$_2$ and SI-NO$_2$ measurements.'

**Line 16-18**: High wind speed and PBL height suppress the fluctuation of NO2? Why? Usually mixing is enhanced by the high wind speed, then the fluctuation becomes larger. → The planetary boundary layer was now replaced by advected marine boundary layer. This is because our study area is coastal area where westerly winds frequently cause TIBL (Thermal Internal Boundary Layer).

**Line 25-27**: This is not the good conclusion after the GMAP 2020 campaign. Everybody knows the difference between column and surface NO2 value. This study can have a merit because the difference between column and surface NO2 was analysed and diagnosed in detail using dense measurement of data. Please make a better statement to underline this merit. ➔ The original paragraph 'The results of this study also indicate that when performing GEMS validation using either PC or SI observations alone, particularly under prevailing local wind meteorological conditions or transport processes', has been replaced by '*The discrepancies suggest that using either PC-NO2 or SI-NO2 observations alone involves a higher possibility of uncertainty under LD conditions or prevailing transport processes*

**Line 39 and whole manuscript**: 'column density' sounds much better that 'column amount'➔ Done.

**Line 39-41**: Is there no equation number? And what is the reference of this equation? ➔ As a reference, Zhao et al. (2019) has been added. Zhao et al. (2019) defined this simple but robust scaling method based on Lamsal et al. (2008) and McLinden et al. (2014). The equation number was not added, because only single equation is used in the MS and no needs to distinguish between multiple equations.

**Line 42-57**: Based on this paragraph, readers expect to see the importance of 'a priori vertical profile' for the accuracy of column data. But that is not the key point of this study right? The main finding of this study is the examination for the relationship between column and surface NO2 in terms of meteorological pattern, acquired by the clustering. Again, the background work suggested in the introduction chapter is not well connected to the key point of this study. ➔ Yes. As noted in the answer to the general comments at the top of this document, MS has been duly treated, underlining that our focus is on the exploring the discrepancies between PI and SI depending on meteorological conditions. In the original MS, our goal description in the paragraph starting at Line 59(starting with 'By contrast') is now looking relatively more emphasized.

**Line 58**: The meaning 'weak vertical profile correlation' is not clear. what is the profile correlation? ➔ This sentence has been now replaced by '*weak correlations between column and surface measurements*' in the revised MS.

**Line 63-64**: The meaning of statement "found that they originated ... from the surface layer" is not clear. How this explains the weak correlation between PC and SI NO2 during the KORUS-AQ? What is different from Wang and Christophere (2003) showing the high correlation in Alabama? ➔We now paraphrased this to '.. and suggested, as a possible reason, the transported non-uniform plumes originated in China, and Seoul hundreds of meters above the ground from the surface layer above the ground' (see Line 57-59) In the revised MS.

**Line 85-86**: 'Low-orbit' and 'geostrationary' cannot be used together. ➔ It has been changed to '*low-orbit or geostationary*' in the revised MS.

**Line 110-123**: I would recommend to focus on the result of GMAP2020 campaign only. The result in GMAP2021can be a PART 2 paper in the future. ➔ As explained earlier, we all removed this part of GMAP-2021 in the Fig. 1, as well as paragraphs described in the main text.

**Line 133**: What is LPS?
➔ In the revised MS, full name + acronyms (in parentheses) for LPS, Met, AQM, and PA were checked and re-described in the text.

**Line 113-114**: Reference? ➔ As the sentences regarding GMAP-2021 campaign information have been deleted, no relevant reference is needed any more.

**Line 158-159**: Cloud cover 0.6 looks a loose criteria. Is PC NO2 quality OK under the cloud cover - 0.5 (50%)? If yes, how is it justified? ➔ Due to the lower data acquisition rate, we employed 0.6. We know that the standard for cloudiness of 0.6 is loose, but it corresponds to the first screening: however, in the subsequent DOAS fitting process, the cases with a large error have been excluded. Therefore, the effect of larger cloudiness has been further removed, and we do not expect the biases to change the major findings of the present study.

**Line 162-163**: I think that this data to 30 October 2021 are not part of GMAP 2021. Again, the solely usage of GMAP 2020 data looks meaningful and better to derive the obvious key message from this work. ➔We duly corrected and removed the sentences relevant to GMAP-2021, hereby only focusing on GMAP-2020, with the only exception with statistics of $PA_4$ that was still mentioned

in Table 1.

**Line 179**: 'NIER-GP2021-002' this format is right as the reference of ACP?

→ (Line 171) We changed to (NIER, 2018), and completed to adjust all of the references, following the acp guideline, as acp-editorial board requested.

**Line 205-222**: The methodology of clustering is not clear. The minimum amount of basic theoretical description is necessary. Here, the usage of XLSTAT software is the only clear part related to the conduction of clustering analysis, which does not look enough. → We shortly explained our implemented k-mean cluster analysis and agglomerative hierarchical cluster approaches, and also cited references (see Line 212-218 in the revised MS).

**Line 213-214**: How did authors hypothesize this? Actually, I can accept this idea, but in the manuscript, the reliable logic / scientific reason is needed to have a hypothesis. → We introduced the relevant references that appears to correlate with our interpretations. (See Line 206-212 in the revised MS)

**Line 239**: Why the correlation is estimated in a 'log' scale? → As indicated in the MS, we presented the results of both linear- and log-scale: a fair logarithmic relationship (R = 0.45), and a relatively lower 1:1 linear relationship (R = 0.41). We guess that log-scale would be more appropriate by direct guessing from the apparent shapes (see the Figures below). See two Figures below.

[Figure]

[Figure]

By comparatively investigating two Figures above, we concurred that Log-fitting is more appropriate, because sharp increase for lower SI-NO2 range (i.e., < ~20 ppb), and almost constant trend for higher SI-NO2 range (i.e., SI-NO2 > ~20 ppb).

Log-scale distribution (high increase for lower SI-NO2, and almost constant for higher SI-NO2) is probably because the Pandora data and SI-NO2 are both on hourly bases (not daily), and SI-NO2 will be compressed (or diluted) directly by shrinking (or developing) PBL, whereas PC-NO2, the total column amount, does not change by PBL. These effects were more pronounced at higher SI-NO2 concentrations. Of course, there should be the exceptional cases that show linear relationships, which have already been presented in Figure 3b, as an example in the current study.

**Line 242-244 + Fig. 3**: This different correlation looks very interesting but there is no explanation about this. Some ideas to descibe this difference should be added here. → We added the description of negatively correlated case (R<0) (Fig. 3b) in the text, such as photochemical process and/or pollutant transport, yielding nonhomogeneous NO2 profile. In the revised MS, we added 'The negative correlation on April 21 (Fig. 3b) implied that the nonhomogeneous NO2 distributions vertically were due to the photochemical process. For example, the decrease in PC-NO2 despite an increase in SI-NO2 might have occurred because NO2 is removed by photochemical loss; it can occur more severely in the upper atmosphere with high OH concentrations. Another possible reason is the occurrence of lifted layers related to pollutant transport, yielding sharp changes in vertical concentration from the surface to the upper layer: The case-specific discussion follows' (see Line 249-255)

**Line 245-247**: This is associated with the colume NO2 or surface NO2? It is not clear. → This is probably for Line 250-260. We now clarified by adding short note such as 'surface' and 'column' in the individual sentences in the revised MS (Line 256-262)

**Line 266-269**: How to find the PBLH using the HYSPLIT simulations? The method is unclear. →This is probably for Line 271-274. We have now clarified that PBLH was simulated by Global Forecast System (GFS), and Lagrangian backward trajectories obtained from Hybrid Single-

Particle Lagrangian Integrated Trajectory (HYSPLIT), installed in GFS system, in the revised MS (Line 279-282)

**Line 270**: Which region relates to this PBLH information? Region is unclear.
→We also clarified the region for the PBLH, by adding a short note 'in Seasan, our study area,' in the revised MS. (Line 282-283)

**Line 278**: What is SBI?→ SBI (Sea Breeze Index) was now mentioned in the caption of Figure 4, and we now defined SBI (with equation number) also in the main text (Line 289-300)

**Line 284-288**: The statement is not well connected to the previous sentence. Please improve.
→We improved the sentences and clarified in the text, such as,

'. Most SBIs in group 1 ranged from 0 to 3, indicating that group 1 corresponded to the dominant local circulation (LD), whereas the SBIs in group 3 had the lowest frequencies comparing 1 and 3, which corresponded to a dominant synoptic-scale circulation (SD). Group 2 can be considered a mixture of local and synoptic-scale circulation (MD)..' (Line 296-300 in the revised MS)

**Line 289-294**: I do not the function of this paragraph. Why do readers think this information related to the result of this study? What can readers know better based on this paragraph?
→ This comment seems to be for Line 294-299. We agree with reviewer's opinion, and removed the whole sentences. We originally tried to understand the vertical profile, fitting to the surface levels, but it does not appear that this is directly relevant to the main point of the current study.

**Line 298-310**: Based on my understanding, authors addressed that the PC-NO2 and SI-NO2 shows good correlation if PBL inside is well-mixed, and this mixing condition is determined by the meterology pattern, therefore we need to consider the meteorological pattern more significantly for the analysis of relationship between column and surface NO2. Am I right? If right, authors need to put the weight more on the role of homogeneity in the PBL to the correlation between column and surface NO2. Was it found before? If yes, discussion with some previous reference is necessary. If not (i.e. this work is the first to show the importance of PBL homegeneity related to the correlation between column and surface NO2), this should be more underlined.

→ This is probably for Line 303-315. We agree and underlined the role of PBL homogeneity in evaluating PC-NO2 and SI-NO2 in more detail, in the revised MS. (see Line 249-255, Line 355-365)

**Line 339-341**: Reference?
→ We added Chong et al. (2018) for MAPS-Seoul, and Herman et. al. (2018) for KORUS-AQ campaign in the revised MS. (Line 349, 352)

**Line 368-370**: How can be the NO2 (short lifetime) transported across the Yellow Sea? It is very debatable. Please add some discussions if authors would state the possibility of NO2 long-range transport with several citations.
→This sentence is for Line 373-378. We added some relevant references and short note for this issue in the revised MS. "This transport of NOx across the region was also discussed and might be particularly high during the winter when the NOx lifetime is relatively longer (Lee et al., 2013; Stohl et al., 2002; Wenig et al., 2003)" (Line 390-392).

**Fig. 1**: Recommend to have the GMAP 2020 information only.
→As described earlier, we have removed the information on GMAP-2021, focusing on GMAP-2020 only.

**Fig. 3b**: This contrast is frequently found or one of correlation is a really irregular one? It requires more and deeper statements. →As stated above, we added the possible reasons chemically (non-linear photochemical reactions) and meteorologically (existence of lifted layer due to transport process) in the revised MS. (see Line 249-255)

**Fig. 5**: Left figure is for the surface NO2, but right figure is for the surface 'delta' NO2, which are different from the absolute value. Please improve the figure caption for better explanation of figures.→We now corrected the Figure, and captions. In the new caption $\triangle NO_2$ was newly defined, such as,

'(b) PC-NO2 vs. Surface $\triangle NO2$ in each meteorological condition over a 1-year period (November 12, 2020–October 30, 2021). Here Surface $\triangle NO2$ = SI-NO2–(30-day moving average) SI-NO2' in the revised MS.

**Fig. 7 and 8**: Figure caption should be corrected. Fig 7 is for flight 5 and 7, but Fig 8 is for flight 1 and 3, so the date and information in detail is different.

→Caption of Fig. 7 has been corrected: (a, b) FL-5 (December 6) and (d, e) FL-7 (December 6 and 9) is correct information. *We thank you for reviewer's thorough comments, and we believe that the comments raised by reviewer has been much more strengthened our MS.*

Reply

**Citation**: https://doi.org/10.5194/acp-2022-170-RC1

---

## Author Comment (AC2)

**Answer to Referee # 2**

(Reviewer's comments in BLACK, authors' reply in BLUE, corresponding modifications in the MS (manuscript) in red)

**General Comments:**

The authors compare in situ measurements and column measurements of $NO_2$. The overall purpose of this comparison is to investigate how well surface measurements can be used for the validation of (GEMS) satellite measurements of $NO_2$. The authors find that the correlation between both data sets depends on meteorological conditions. They define specific classes based on different slopes and correlation coefficients. They explain the different relationships making also use of additional measurements, e.g. vertical profiles from aircraft measurements.

Their main conclusion is that ,caution is required when performing GEMS validation using either PC or SI observations alone, particularly under prevailing local wind meteorological conditions or transport processes.' While I agree that caution is required if only surface observations are available, I disagree that caution is required if tropospheric column amount measurements from independent sources are available. Overall, this is a very useful study and I recommend publication in ACP after major revisions.

→ We appreciate the reviewer's points. GMAP campaign has been actually started in2020 for multi-purpose, including GEMS validation. In the next year 2021, GMAP-2021 campaign was carried out based on more multi-perspective observation instruments from the ground to the space for GEMS validation such as MAXDOAS, Car-DOAS, GCAS, and Pandora data. However, in this MS, regarding GEMS validation, we carefully checked and removed direct connection between GEMS validation and GMAP campaign. We now believe the MS was revised in accordance with reviewer's comments below, and, by reflected the reviewer's suggestions and comments, as bellow.

**Specific Comments:**

**Major points:**

1) As mentioned above, I disagree with the authors that caution is required if tropospheric column amount measurements from independent sources are available. In contrast, such independent data sets (e.g. from MAX-DOAS observations) are a very good source for satellite observations of tropospheric $NO_2$ columns. I suggest to remove this statement from the abstract and from other parts of the paper. → We all agree with reviewer's points. We removed this issue In the revised MS, and highlighted the analysis of relationship between PC-NO2 and SI-NO2, rather than association directly with the GEMS satellite validation. As reviewer recommended, we have all removed the sentences relevant to GEMS validation (over the abstract and the main text as well), such as Line 25-27, 98-100, 123-124, 211-213, 391-393, and 526-528 in the original MS)

2) From Pandora, also tropospheric profiles and tropospheric VCDs are available. These quantities are much better suited for a correlation analysis than the total $NO_2$ columns. It is not clear to me why the authors chose the total VCDs. I suggest that the correlation analysis should be extended (or replaced) using tropospheric VCDs and surface concentrations from pandora measurements.

→We checked further on this point and discussed with co-authors, such as the differences between tropospheric-VCD vs. total-VCD, and we concurred the following points.

In Seosan, we confirmed that the tropospheric-VCD has a high correlation with the total-VCD, mainly because the stratospheric column changes little in space and time at the local scale. Therefore, we guess that it is highly likely that the results from tropospheric-VCD would be almost the same (or similar) as our total-VCD-employed results. However, regarding the data uncertainty, we found out that the quality of tropospheric VCD provided by PGN in Seosan is still low, and some tropospheric VCDs are found to be even larger than the total VCD in some cases. In this background, we thought that it is believed to be still too early to use tropospheric-VCD in our study. Furthermore, PGN does not provide profile data of NO2, and the atmospheric aerosol-loading is significant over the East Asia, and accordingly it is highly likely to cause considerable uncertainty in the NO2 profile calculation if it is not properly treated. As an example of uncertainty, ground concentration level calculated by Pandora shows, when compared to SI-NO2, the lower correlation ($R^2$<0.2) was found and the tendency to underestimate (slope<10%) is highly likely, as seen below. In this situation, total-VCD would be still useful to employ for our

study. However, as reviewer pointed out, it seems unreasonable to connect our findings to connect 'GEMS validation purpose', and therefore removed those through the MS, as mentioned above.

[Figure]

3) I suggest to extend the correlation analysis by looking how the correlation between PC or SI observations depends on the time of the day. This would be a very valuable addition. We can expect that the correlation changes with time, because also the vertical mixing and the photolysis rate changes with time.

→ We carried out further analysis to check this reviewer's point: tropospheric-VCD vs. total-VCD, and added the description and add a supplementary Figure (Figure S1, see below) in the revised MS.

[Figure]

**Figure S1.** Correlation coefficients between PC-NO2 vs. SI-NO2 at (a)0900~1200 LST and (b)1400~1700LST, observed at Seosan during GMAP-2020 campaign.

(In the main text)

  "In this study, we extended the correlation analysis, and investigated the correlation between PC-NO2 and SI-NO2 measurements on an hourly basis. The results showed the correlation between PC-NO2 and SI-NO2 have a lower correlation in the morning, and a higher correlation in the afternoon (Fig. S1). The median correlation coefficients for three LD, MD, and SD meteorological conditions were -0.71, 0.18, and 0.22 in the morning (0900~1200LST), and 0.84, 0.77, and 0.79 in the afternoon (1200~1400LST), respectively. These can be interpreted from the PBL development. SI-NO2 decreases in the morning due to the rapid PBL growth, while PC-NO2 increases due to the accumulation of NO2 in the atmosphere. However, there is very little change in PBL in the afternoon, and PC-NO2 and SI-NO2 show similar changes, yielding a positive correlation each other. Our study is limited to GMAP campaign period; thus more detailed interpretations would be needed to infer more plausible causes." (see Line 355-365 in the revised MS).

4) line 241: You write: ,These hourly data exhibited a fair logarithmic relationship (R = 0.45),'

(Why) do you calculate the logartithmic relationship? Please clarify→ This is also pointed out by another reviewer. We guess that log-scale would be more appropriate by direct guessing from the apparent shapes (see the Figures below). See two Figures below.

[Figure]

[Figure]

Simply by investigating two Figures above, it appears that Log-fitting is more appropriate, because sharp increase for lower SI-NO2 range (i.e., < ~20ppb), and almost constant trend for higher SI-NO2 range (i.e., SI-NO2 > ~20ppb). Log-scale distribution (high increase for lower SI-NO2, and almost constant for higher SI-NO2) is probably because the Pandora data and SI-NO2 are both on hourly bases (not daily), and SI-NO2 will be compressed (or diluted) directly by shrinking (or developing) PBL, whereas PC-NO2, the total column amount, does not change by PBL. These effects were more pronounced at higher SI-NO2 concentrations. Of course, there should be the exceptional cases that show linear relationships, which have already been presented in Figure 3b, as an example in the current study.

**Minor points:**

Line 159: It is not clear how direct light measurements can be performed under partly cloud-covered skies. Please clarify.

→ Clouds increase the PC-NO2 noise and lower precision of measurement. In the case that cloud cover is small (thin or moderate), PC-NO2 can be well estimated (Herman et al., 2009, JGR). This is because, when direct light passes through a thin cloud, the sunlight reaching Pandora is constantly reduced in proportion to the wavelength; therefore, this linearity makes it possible to retrieve the PC-NO2 density. However, in the case of thick cloud, however, the proportion of scattered light becomes greater compared to direct light, and the retrieval error increases significantly. Therefore, we limited the data used in this study to that with an observed uncertainty of 0.01 DU (or less). Probably reviewers (and Journal readers also) would know the relation of clouds vs. PC-NO2-uncertainty, and thus we concluded that it is not needed to put details in the main text; only added the reference in the revised MS.

Line 254: how can you expect a constant value? The stratospheric $NO_2$ amount varies with season (and time of the day). Please clarify. → We then clarified the period of our estimation by adding 'during the GMAP campaign period', and specified the application period. As a reference, we estimated monthly variations of the stratospheric NO2 amount from both TROPOMI through

2020/Nov. ~ 2021/Oct. (see the Table below). The results showed that stratospheric NO2 is low in winter and high in late spring. During the GMAP-2020 (Nov. Dec. and Jan), stratospheric NO2 was found to be 0.086 DU(=[0.084+0.09+0.085]/3DU) within the analysis period, well comparable with our results. The result also showed that the average is 0.1 DU, while the standard deviation is 0.02 DU, and the sigma/mean is within about 17.4%, implying that the fluctuation range is not large. Therefore, we did not mention further or add further information about this in the revised MS.

| Month | Stratospheric NO2 (DU) |
|---|---|
| 11 | 0.084 |
| 12 | 0.09 |
| 1 | 0.085 |
| 2 | 0.095 |
| 3 | 0.111 |
| 4 | 0.119 |
| 5 | 0.137 |
| 6 | 0.134 |
| 7 | 0.132 |
| 8 | 0.127 |
| 9 | 0.118 |
| 10 | 0.107 |
| Mean | 0.111583 |
| Standard Deviation | 0.019374 |
| Sigma/Mean | 17.3629 |

Line 361: You write: ,All observed NO2 profiles shown in Fig. 7 appeared to have generally exponential curves...'I think you cannot conclude this because there is not measurement between the surface and 500m. Please clarify.

→ This was also pointed out by another reviewer, and we deleted all this sentence in the revised MS.

*We appreciate the reviewer's insightful comments that are believed to have much more strengthened our MS.*

---

## Author Response (AR2)

**Answer to Referee # 1 (Previously Referee #2)**

**Key Comments:**

The authors have well addressed my major points #1 and #3. Many thanks! They also provided a reasonable explanation to my major point #2. Nevertheless, here I strongly suggest that the authors add a short explanation (like their reply to my major point #2) to the paper why not the tropospheric NO2 VCD, but the total NO2 VCD is used in the study. Concerning my major point #4 I suggest that the authors use a linear correlation. I don't see a justification to use a logarithmic function.

**Response to (Point #2)**

→ We added a short description on the revised MS (L145 – 151) why not use tropospheric VCD and use total VCD on the other hand, such as

*"From retrieved Pandora measurements, tropospheric and total (=tropospheric + stratospheric) vertical column densities are both available. However, it should be noted that appreciable uncertainties cannot be neglected in the tropospheric NO2 profiles obtained from Pandora instruments, particularly for the high aerosol-loading areas such as East Asia. In this background, we used total vertical column densities in the present study, and also confirmed that they have a high correlation with the tropospheric column densities observed in our study period with little change in stratospheric column density in space and time at the local scale."* See Line 145 – 151 in the revised MS.

**Response to (Point #4)**

→ We replaced the Log-fitting by 1:1 linear fitting (see newly plotted Fig. 3 below), and have all REMOVED the Log-fitting and its relevant descriptions, such as correlation coefficients estimated from logarithmic function (L249-251, and L565-566 in original MS) and descriptions on y-interceptor from Log-function (L265-271 and L567-569 in original MS).

[Figure]

**Figure 3.** a) Pandora column (PC) NO$_2$ measurements as a function of surface *in situ* (SI) NO$_2$ observations at Pandora sites PA$_1$–PA$_3$ during the GMAP-2020 campaign and PA$_4$ during a 1-year period. A 1:1 linear regression model was used to evaluate the relationship between PC and SI measurements (black line). (b) Sample scatterplots of PC-NO$_2$ and SI-NO$_2$ for February 24 (red) and April 21 (blue), 2021.

*We believe that revised manuscript has been much more strengthened, and we appreciate the reviewer's insightful comments.*